# Uncovering temporal structure in hippocampal output patterns

**Kourosh Maboudi[1,2†], Etienne Ackermann[3†], Laurel Watkins de Jong[1,2], Brad E Pfeiffer[4], David Foster[5,6], Kamran Diba[1,2‡*], Caleb Kemere[3‡*]**

[1]Departmentof Anesthesiology, University of Michigan, Ann Arbor, United States; [2]Department of Psychology, University of Wisconsin-Milwaukee, Milwaukee, United States; [3]Department of Electrical and Computer Engineering, Rice University, Houston, United States; [4]Department of Neuroscience, University of Texas Southwestern, Dallas, United States; [5]Department of Psychology, University of California, Berkeley, Berkeley, United States; [6]Helen Wills Neuroscience Institute, University of California, Berkeley, Berkeley, United States

**Abstract** Place cell activity of hippocampal pyramidal cells has been described as the cognitive substrate of spatial memory. Replay is observed during hippocampal sharp-wave-ripple-associated population burst events (PBEs) and is critical for consolidation and recall-guided behaviors. PBE activity has historically been analyzed as a phenomenon subordinate to the place code. Here, we use hidden Markov models to study PBEs observed in rats during exploration of both linear mazes and open fields. We demonstrate that estimated models are consistent with a spatial map of the environment, and can even decode animals' positions during behavior. Moreover, we demonstrate the model can be used to identify hippocampal replay without recourse to the place code, using only PBE model congruence. These results suggest that downstream regions may rely on PBEs to provide a substrate for memory. Additionally, by forming models independent of animal behavior, we lay the groundwork for studies of non-spatial memory.
DOI: https://doi.org/10.7554/eLife.34467.001

*For correspondence:
kdiba@umich.edu (KD);
caleb.kemere@rice.edu (CK)

[†]These authors contributed equally to this work
[‡]These authors also contributed equally to this work

**Competing interests:** The authors declare that no competing interests exist.

## Introduction

Large populations of neurons fire in tandem during hippocampal sharp-waves and their accompanying CA1 layer ripple oscillations (*Buzsáki, 1986*). By now, multiple studies have shown that during many sharp-wave ripple-associated PBEs, hippocampal 'place cells' (*O'Keefe and O'Keefe, 1976*) fire in compressed sequences that reflect the firing order determined by the sequential locations of their individual place fields (*Diba and Buzsáki, 2007*; *Foster and Wilson, 2006*; *Lee and Wilson, 2002*; *Nádasdy et al., 1999*). While the firing patterns during active exploration are considered to represent the brain's global positioning system and provide a substrate for spatial and episodic memory, instead it is the synchronized activity during PBEs that is most likely to affect cortical activity beyond the hippocampus (*Buzsáki, 1989*; *Carr et al., 2011*; *Diekelmann and Born, 2010*; *Siapas and Wilson, 1998*). Likewise, widespread activity modulation is seen throughout the brain following these sharp-wave ripple population bursts (*Logothetis et al., 2012*).

The literature on PBEs has largely focused on developing templates of firing patterns during active behavior and evaluating the extent to which these templates' patterns are reprised during subsequent PBE. But what if the fundamental mode of the hippocampus is not the re-expression of place fields, but rather the PBE sequences during SWR? PBE sequences are enhanced during exploration of novel environments (*Cheng and Frank, 2008*; *Foster and Wilson, 2006*), they presage learning-related changes in place fields (*Dupret et al., 2010*), and appear to be critical to task learning (*Ego-Stengel and Wilson, 2010*; *Girardeau et al., 2009*; *Jadhav et al., 2012*). Here, we

examine the information provided by CA1 and CA3 pyramidal neurons, the output nodes of the hippocampus, through the looking glass of PBE firing patterns.

We developed a technique to build models of PBE sequences strictly outside of active exploration and independent of place fields and demonstrate that this nevertheless allows us to uncover spatial maps. Furthermore, these models can be used to detect congruent events that are consistent with replay but without any explicit place cell template. Our technique therefore provides new possibilities for evaluating hippocampal output patterns in single-trial and other fast learning paradigms, where a reliable sequential template pattern is not readily available. Overall, our work suggests that a sequence-first approach can provide an alternative view of hippocampal activity that may shed new light on how memories are formed, stored, and recalled.

## Results

### Awake population burst events

We began by analyzing the activity of large numbers of individual neurons in areas CA1 and CA3 of the dorsal hippocampus as rats navigated linear mazes for water reward (linear track: $n = 3$ rats, $m = 18$ sessions; previously used by [*Diba and Buzsáki, 2007*]). Using pooled multiunit activity, we detected PBEs during which many neurons were simultaneously active. The majority of these events occurred when animals paused running (speed <5 cm/s, corresponding to $54.0\% \pm 20.1\%$ sd of events) to obtain reward, groom, or survey their surroundings (*Buzsáki et al., 1983*), and were accompanied by SWR complexes, distinguished by a burst of oscillatory activity in the 150–250 Hz band of the CA1 LFP. Because we are interested in understandinginternally generated activity during PBEs, we included only these periods without active behavior, ensuring that theta sequences would not bias our results. While we identified active behavior using a speed criterion, we found similar results when we instead used a theta-state detection approach (not shown). We did not add any other restrictions on behavior, LFP, or the participation of place cells. We found that inactive PBEs occupied an average of 1.8% of the periods during which animals were on the linear track ($16.9 \pm 15.1$ s of $832.6 \pm 390.5$ s). In comparison, classical Bayesian approaches to understand PBE activity require the 34.8% of time animals are running (speed >10 cm/s) on the track ($254.4 \pm 106.6$ s of $832.6 \pm 390.5$ s) to build models of place fields.

### Learning hidden Markov models from PBE data

Activity during PBEs is widely understood to be internally-generated in the hippocampal-entorhinal formation, and likely to affect neuronal firing in downstream regions (*Buzsáki, 1989*; *Chrobak and Buzsáki, 1996*; *Logothetis et al., 2012*; *Yamamoto and Tonegawa, 2017*). Given the prevalence of PBEs during an animal's early experience, we hypothesized that the neural activity during these events would be sufficient to train a machine learning model of sequential patterns—a hidden Markov model—and that thismodel would capture the relevant spatial information encoded in the hippocampus independent of exploration itself.

Hidden Markov models have been very fruitfully used to understand sequentially structured data in a variety of contexts. A hidden Markov model captures information about data in two ways. First, it clusters observations into groups ('states') with shared patterns. In our case, this corresponds to finding time bins in which the same sets of neurons are co-active. This is equivalent to reducing the dimension of the ensemble observations into a discretized latent space or manifold. Second, it models the dynamics of state transitions. This model is Markovian because it is assumed that the probability to transition to the next state only depends on the current state. Critically, these operations of clustering and sequence modeling are jointly optimized, allowing the structure of ensemble firing corresponding to each of the final states to combine information over many observations. Given the role of the hippocampus in memory, in our HMMs, the unobserved latent variable presumably corresponds to the temporal evolution of a memory trace that is represented by co-active ensembles of CA1 and CA3 neurons. The full model will correspond to the structure which connects all the memory traces activated during PBEs.

The parameters of our model that are fit to data include the observation model (the cluster descriptions, or predicted activity of each excitatory neuron within the CA1/CA3 ensemble for a given state), the state transition model (the probability that the CA1/CA3 ensemble will transition

from a start state to a destination state in the next time bin), and the initial state distribution (the probability for sequences to start in each given state). In prior work using HMMs to model neural activity, a variety of statistical distributions have been used to characterize ensemble firing during a specific state (the observation model, (*Chen and Wilson, 2017*; *Chen et al., 2012*; *Chen et al., 2014*; *Deppisch et al., 1994*; *Kemere et al., 2008*; *Radons et al., 1994*). We opted for the Poisson distribution to minimize the number of parameters per state and per neuron (see Materials and methods). We used the standard iterative EM algorithm (*Rabiner, 1989*) to learn the parameters of an HMM from binned PBE data (20 ms bins). *Figure 1* depicts the resultant state transition matrix and observation model for an example linear-track session.

Using separate training- and test-datasets (cross-validation) mitigates over-fitting to training data, but it is still possible for the cross-validated goodness-of-fit to increase with training without any underlying dynamics, e.g., if groups of neurons tend to activate in a correlated fashion. Does the model we have learned reflect underlying sequential structure of memory traces beyond pairwise co-firing? To answer this question, we cross-validated the model against both real 'test' data and against surrogate 'test' data derived from shuffling each PBE in two ways: one in which the binned spiking activity was circularly permuted across time for each neuron independently of the other neurons ('temporal shuffle', which removes co-activation), and one in which the order of the binned data was scrambled coherently across all neurons ('time-swap', which maintains co-activation). Note that the second shuffle preserves pairwise correlations while removing the order of any sequential

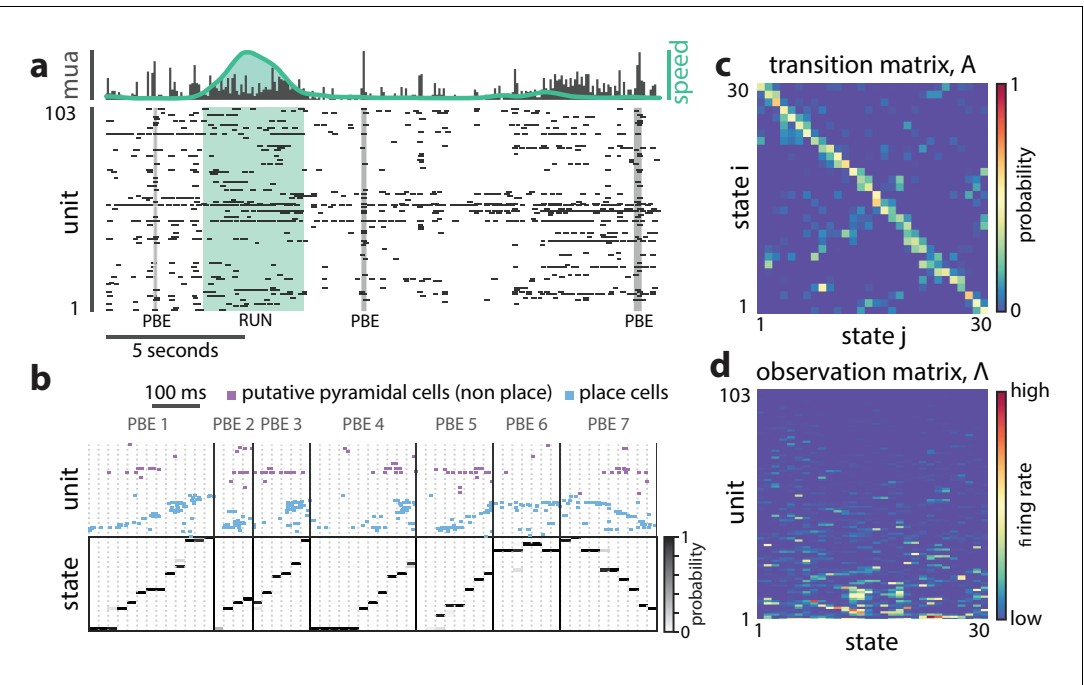

**Figure 1.** A hidden Markov model of ensemble activity during PBEs. A hidden Markov model of ensemble activity during PBEs. (**a**) Examples of three PBEs and a run epoch. (**b**) Spikes during seven example PBEs (top) and their associated (30 state HMM-decoded) latent space distributions (bottom). The place cells are ordered by their place fields on the track, whereas the non-place cells are unordered. The latent states are ordered according to the peak densities of the lsPFs (lsPFs, see Materials and methods). (**c**) The transition matrix models the dynamics of the unobserved internally-generated state. The sparsity and banded-diagonal shape are suggestive of sequential dynamics. (**d**) The observation model of our HMM is a set of Poisson probability distributions (one for each neuron) for each hidden state. Looking across columns (states), the mean firing rate is typically elevated for only a few of the neurons and individual neurons have elevated firing rates for only a few states.

DOI: https://doi.org/10.7554/eLife.34467.002

The following figure supplement is available for figure 1:

**Figure supplement 1.** Hidden Markov models capture state dynamics beyond pairwise co-firing.
DOI: https://doi.org/10.7554/eLife.34467.003

patterns that might be present. Using five-fold cross-validation, we compared learned models against both actual and surrogate test data and found that the model likelihood was significantly greater for real data (vs. temporal shuffle, $p<0.001$, vs. time-swap, $p<0.001$, $n = 18$ sessions, Wilcoxon signed-rank test, *Figure 1—figure supplement 1*).

## What do the learned model parameters tell us about PBEs?

To begin to understand what structure we learn from PBE activity, we compared our HMMs (trained on real data) against models trained on multiple different surrogate datasets (*Figure 2a,b*). These surrogate datasets were obtained from actual data following: (1) temporal shuffles and (2) time-swaps, as above, and (3) by producing a surrogate PBE from independent Poisson simulations according to each unit's mean firing rate within the original PBEs. First, we investigated the sparsity of the transition matrices using the Gini coefficient (see Materials and methods and *Figure 2—figure supplement 1*). A higher Gini coefficient corresponds to higher sparsity. Strikingly, the actual data yielded models in which the state transition matrix was sparser than in each of the surrogate counterparts ($p<0.001$, *Figure 2c*), reflecting that each state transitions only to a few other states. Thus, intricate yet reliable details are captured by the HMMs. Next, we quantified the sparsity of the

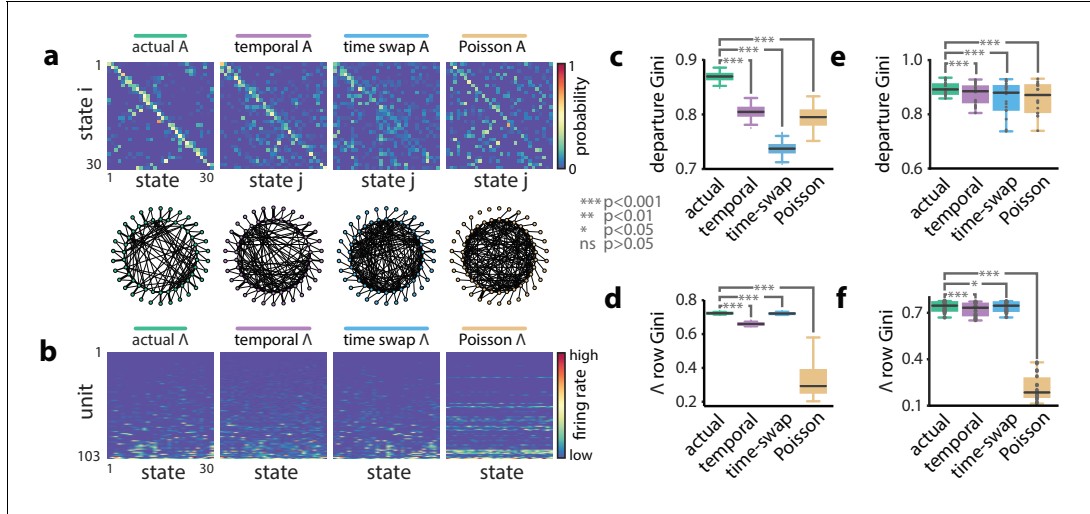

**Figure 2.** A hidden Markov model of ensemble activity during population burst events. Models of PBE activity are sparse. We trained HMMs on neural activity during PBEs (in 20 ms bins), as well as on surrogate transformations of those PBEs. (a) (top) The transition matrices for the actual and surrogate PBE models with states ordered to maximize the transition probability from state $i$ to state $i + 1$. (bottom) Undirected connectivity graphs corresponding to the transition matrices. The nodes correspond to states (progressing clockwise, starting at the top). The weights of the edges are proportional to the transition probabilities between the nodes (states). The transition probabilities from state i to every other state except $i + 1$ are shown in the interior of the graph, whereas for clarity, transition probabilities from state $i$ to itself, as well as to neighboring state $i + 1$ are shown between the inner and outer rings of nodes (the nodes on the inner and outer rings represent the same states). (b) The observation matrices for actual and surrogate PBE models show the mean firing rate for neurons in each state. For visualization, neurons are ordered by their firing rates. (c) We quantified the sparsity of transitions from one state to all other states using the Gini coefficient of rows of the transition matrix for the example session in (a). Actual data yielded sparser transition matrices than shuffles. (d) The observation models—each neuron's expected activity for each state—learned from actual data for the example session are significantly sparser than those learned after shuffling. This implies that as the hippocampus evolves through the learned latent space, each neuron is active during only a few states. (e) Summary of transition matrix sparsity and f. Observation model sparsity with corresponding shuffle data pooled over all sessions/animals. (***: $p<0.001$, *: $p<0.05$; single session comparisons: $n = 250$ realizations, Welch's t-test; aggregated comparisons - $n = 18$ sessions, Wilcoxon signed-rank test).

DOI: https://doi.org/10.7554/eLife.34467.004

The following figure supplements are available for figure 2:

**Figure supplement 1.** PBE model states typically only transition to a few other states.

DOI: https://doi.org/10.7554/eLife.34467.005

**Figure supplement 2.** Each neuron is active in only a few model states.

DOI: https://doi.org/10.7554/eLife.34467.006

**Figure supplement 3.** The sparse transitions integrate into long sequences through the state space.

DOI: https://doi.org/10.7554/eLife.34467.007

observation model. We found that actual data yielded mean firing rates which were highly sparse (*Figure 2d*), indicating that individual neurons were likely to be active during only a small fraction of the states. Using a graph search algorithm (see Materials and methods), we simulated paths through state space generated by these transition matrices, and found that this increased sparsity accompanied longer trajectories (*Figure 2—figure supplement 3*) through the state space of the model. Thus, the state transition matrices we learn are suggestive of dynamics in which each sparse state is preceded and followed by only a few other, in turn, sparse states, providing long sequential paths through state space-consistent with spatial relationships in the environment in which the animal was behaving, but generated from PBEs. The increased sparsity of the observation model and transition matrix in the example session was representative of a significant increase over all remaining sessions ($p<0.05$, $n = 18$ sessions, Wilcoxon signed-rank tests, *Figure 2e,f*).

These observations indicate that PBEs inform an HMM about extant spatial relationships within the environment. So, next we asked how the firing patterns of neurons during actual behavior project into the learned latent spaces. To observe the evolution of the latent states during behavior, we used our model to determine the most likely sequence of latent states corresponding to decode the neural activity observed in 100 ms bins during epochs that displayed strong theta oscillations (exclusive of PBEs) when rats were running (speed >10 cm/s; see Materials and methods). If the learned model was distinct from ensemble patterns during behavior, we might expect the resulting state space probability distributions at each point in time to be randomly spread among multiple states. Instead, we found distributions that resembled sequential trajectories through the latent space (*Figure 3a*) in parallel with the physical trajectories made by the animal along the track, further

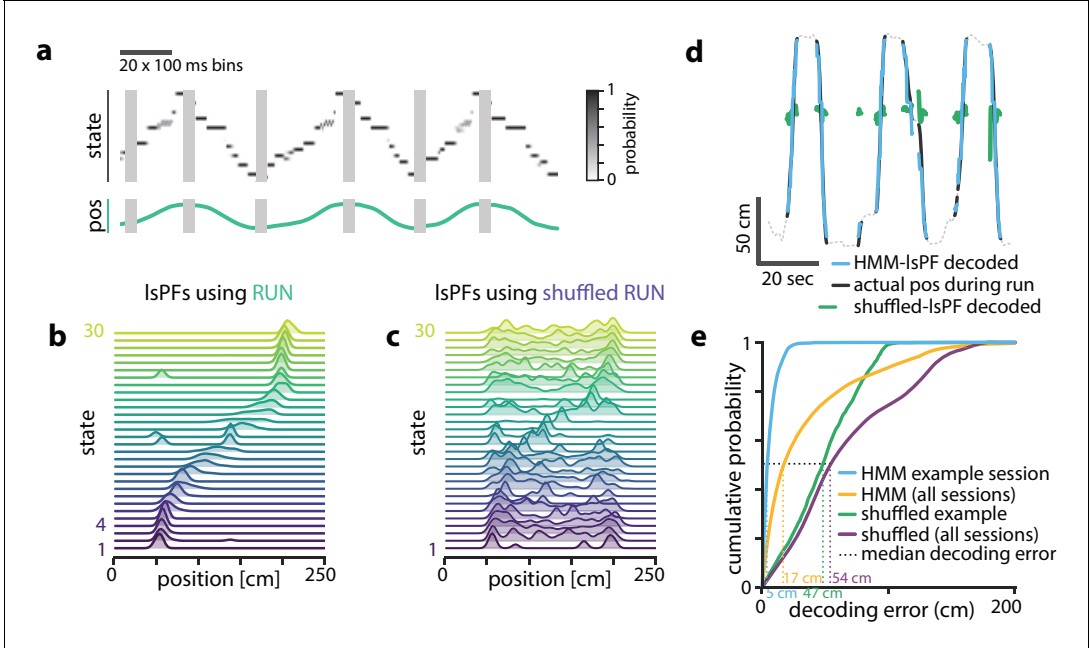

**Figure 3.** Latent states capture positional code. Latent states capture positional code. (a) Using the model parameters estimated from PBEs, we decoded latent state probabilities from neural activity during periods when the animal was running. An example shows the trajectory of the decoded latent state probabilities during six runs across the track. (b) Mapping latent state probabilities to associated animal positions yields latent-state place fields (lsPFs) which describe the probability of each state for positions along the track. (c) Shuffling the position associations yields uninformative state mappings. (d) For an example session, position decoding during run periods through the latent space gives significantly better accuracy than decoding using the shuffled tuning curves. The dotted line shows the animal's position during intervening non run periods. (e) The distribution of position decoding accuracy over all sessions ($n = 18$) was significantly greater than chance. ($p<0.001$).
DOI: https://doi.org/10.7554/eLife.34467.008

The following figure supplement is available for figure 3:

**Figure supplement 1.** Latent states capture positional code over wide range of model parameters.
DOI: https://doi.org/10.7554/eLife.34467.009

demonstrating that the latent state dynamics learned from PBEs corresponds to an internalized model of physical space.

To better understand the relationship between the latent space and physical space, we used the latent state trajectories decoded during running to form an estimate of the likelihood of each state as a function of location on the track (see Materials and methods). These lsPF 'lsPFs' (lsPF, *Figure 3b*) in many ways resembled neuronal place fields and similarly tiled the extent of the track. This spatial localization went away when we re-estimated the lsPF with shuffled positions (*Figure 3c*). To quantify how informative the latent states were about position, we used the lsPF to map decoded state sequences to position during running periods (*Figure 3d*). In our example session, decoding through the latent space resulted in a median accuracy of 5 cm, significantly greater than the 47 cm obtained from shuffled lsPF (*p*<0.001, Wilcoxon signed-rank test, *Figure 3d*). When we evaluated decoding error over our full set of sessions, we observed a similar result (*p*<0.001, Wilcoxon signed-rank test, *Figure 3e*, *Figure 3—figure supplement 1*). As our method required discretizing the state space, a potential caveat is that the number of latent states is a relevant parameter, which we arbitrarily chose to be 30. However, latent-state place fields were informative of position over a wide range of values of this parameter (*Figure 3—figure supplement 1*). Note that decoding into the latent space and then mapping to position resulted in slightly higher error than simply performing Bayesian decoding on the neural activity during behavior. This suggests that the latent space we learn from PBEs may not capture all the information about space that is present in hippocampal activity during behavior, though this may also reflect the limited number of PBEs from which we can learn.

## HMM-congruent PBEs capture sequence replay

We and others have previously described how the pattern of place cell firing during many PBEs recapitulates the order in which they are active when animals run on the track (*Figure 4a*). We employed the versatile and widely-used Bayesian decoding method to ascribe a replay score to sequential patterns during PBEs. Briefly, for each PBE, we used place-field maps to estimate a spatial trajectory (an *a posteriori* distribution of positions) in 20 ms bins. We generated surrogate data via a column-cycle shuffle (i.e., a circular shift across positions for each time bin [*Davidson et al., 2009*]) of the *a*

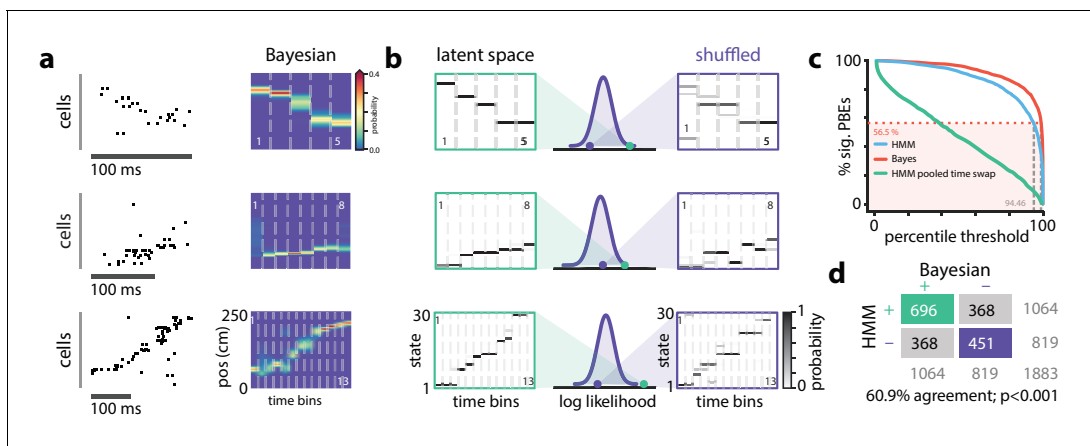

**Figure 4.** Replay Events Can Be Detected Via HMM Congruence. (**a**) Example PBEs decoded to position using Bayesian decoding. (**b**) (left) Same examples decoded to the latent space using the learned HMM. (right) Examples decoded after shuffling the transition matrix, and (middle) the sequence likelihood using actual and shuffled models. (**c**) Effect of significance threshold on the fraction of events identified as replay using Bayesian decoding and model congruent events using the HMM approach. (**d**) Comparing Bayesian and model-congruence approaches for all PBEs recorded, we find statistically significant agreement in event identification (60.9% agreement, *n* = 1883 events from 18 sessions, *p*<0.001, Fisher's exact test two sided).

DOI: https://doi.org/10.7554/eLife.34467.010

The following figure supplement is available for figure 4:

**Figure supplement 1.** Number of significant PBEs.

DOI: https://doi.org/10.7554/eLife.34467.011

*posteriori* distributions during PBEs. The real and surrogate trajectories were scored (see Materials and methods), and we defined replay events as those for which the score of the actual trajectory was larger than a threshold fraction of the null distribution generated by the surrogate scores. Using this approach, we found that 57% of PBE (1064 of 1883) were identified as replay beyond a threshold of 99% (median across datasets 54.2%, $\mathrm{interquartile\ range} = 32.8{-}61.0\%$, *Figure 4—figure supplement 1*). Thus, as has been reported many times (*Davidson et al., 2009*; *Diba and Buzsáki, 2007*; *Foster and Wilson, 2006*; *Karlsson and Frank, 2009*), only a fraction of PBEs (but many more than expected by chance) represent statistically significant replay. Given that we use all PBEs for model learning and our models capture the structure of the environment and the patterns expressed by place cells during exploration, we were interested in understanding whether we could also use our latent-space models to find these replay events. Indeed, for many events when we decode trajectories through state space, they resemble the sequential patterns observed when we decode position using Bayesian techniques and the place cell map (*Figure 4b*, left). However, given previous evidence for replay of environments not recently experienced (*Gupta et al., 2010*; *Karlsson and Frank, 2009*), we hypothesized that some PBEs might contain ensemble neural activity which is unstructured and thus unrelated to the learned model, and that these would correspond to the 'non-replay' events found using traditional methods.

To assess how well the pattern of ensemble activity during individual PBEs related to the overall state-space model learned from PBE activity ('congruence'), we developed a statistical approach for identifying the subset of strongly structured PBEs. Specifically, rather than comparing real and surrogate PBEs, we compared the goodness-of-fit for each event to a null distribution generated via a computationally-efficient manipulation of the transition matrix of the model (*Figure 4b*); we row-wise shuffled the non-diagonal elements of the transition matrix to assess whether an individual PBEs is a more ordered sequence through state space than would be expected by chance. Maintaining the diagonal avoids identifying as different from chance sequences which consist of few repeated states, marked by transitions between state $i$ and itself. As described above, the fraction of events identified as replay using Bayesian decoding is strongly tied to how the null-distribution is generated (i.e., what shuffle is used), some secondary criteria (e.g., number of active cells, unit cluster quality, peak firing rate, trajectory 'jumps', etc.), and the value of the significance threshold arbitrarily chosen to be 90%, 95%, or 99% of shuffles in different reports. When we combined across datasets, we found that our transition matrix shuffle yielded a null distribution for which a 99% confidence interval identified slightly fewer PBEs as significant than the column-cycle shuffle did for Bayesian decoding (*Figure 4c*). To make a principled comparison of Bayesian- and HMM-based replay detection schemes, we fixed the Bayesian-based significance threshold at 99% but selected the significance threshold for the HMM-congruence null distribution so that the fraction of replay events detected would be the same between the two schemes. Following this approach, we found that model-congruent/incongruent PBEs largely overlapped with the replay/non-replay events detected using Bayesian decoding of the place cell map (*Figure 4d*). Thus, using only the neural activity during PBEs, without access to any place cell activity, we are remarkably able to detect the sequential patterns typically described as 'replay' based only on their consistency with the structure of other PBE activity.

There were, however, also differences between the Bayesian and HMM-congruent approaches, including events that reached significance in one but not the other formalism. We wanted to understand where and why these approaches differed in identifying significant sequences. When we examined individual PBEs, we found sequences for which both Bayesian and model-congruence replay detection approaches appeared to malfunction (*Figure 5a*). This was not a failure of the choice of significance threshold, as for both techniques we found what appeared to be false-negatives (patterns which looked like replay but were not significant) as well as false-positives (patterns which looked noisy but were identified as replay). Thus, in order to quantitatively compare the two approaches, we asked eight humans to visually examine all the PBEs in our database. They were instructed to label as replay PBEs in which the animal's Bayesian decoded position translated sequentially without big jumps (*Silva et al., 2015*; see Materials and methods).

We marked each event as a 'true' community replay if it was identified by a majority of scorers (six individuals scored $n = 1883$ events, two individuals scored a subset of $n = 1423$ events, individual scores are shown in *Figure 5—figure supplement 1*). We calculated an ROC curve which compared the rate of true positive and false positive detections as the significance thresholds for Bayesian and

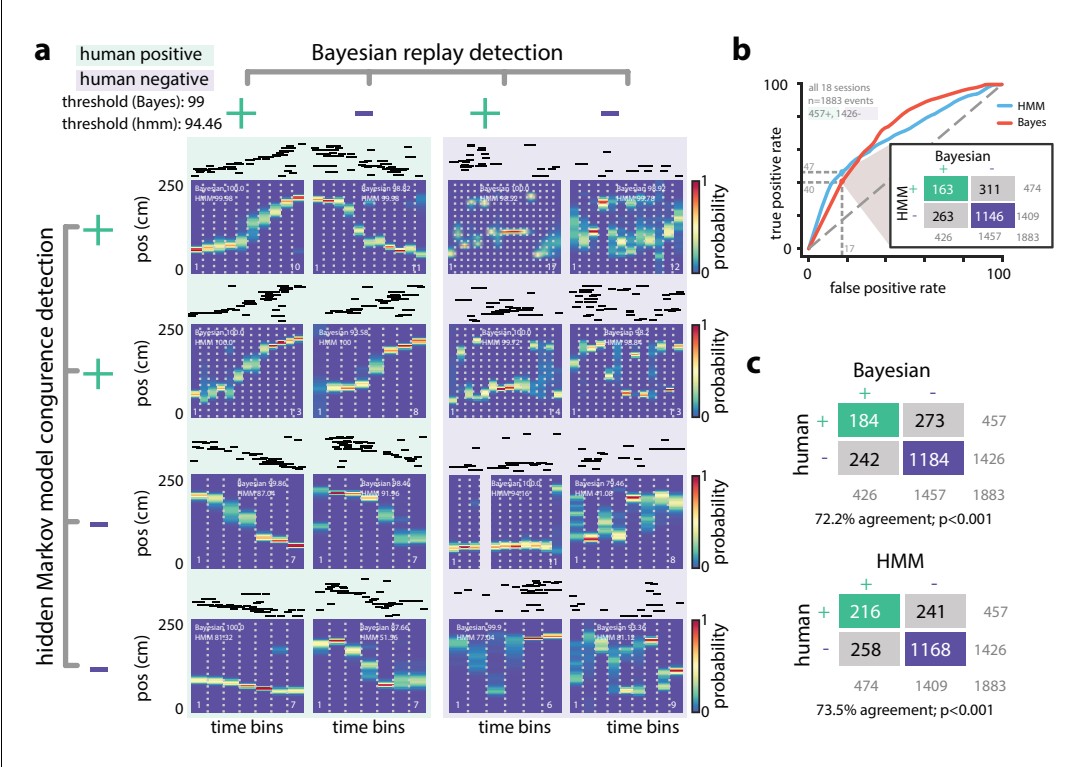

**Figure 5.** Comparing HMM congruence and Bayesian decoding in replay detection. (a) Eight examples from one session show that Bayesian decoding and HMM model-congruence can differ in labeling of significant replay events. For each event, spike rasters (ordered by the location of each neuron's place field) and the Bayesian decoded trajectory are shown. '+' ('-') label corresponds to significant (insignificant) events. (left) Both methods can fail to label events that appear to be sequential as replay and (right) label events replay that appear non-sequential. (b) We recruited human scorers to visually inspect Bayesian decoded spike trains and identify putative sequential replay events. Using their identifications as labels, we can define an ROC curve for both Bayesian and HMM model-congruence which shows how detection performance changes as the significance threshold is varied. (inset) Human scorers identify 24% of PBEs as replay. Setting thresholds to match this value results in agreement of 70% between Bayesian and HMM model-congruence. (c) Using the same thresholds, we find ≈70% agreement between algorithmic and human replay identification. (All comparison matrices, $p<0.001$, Fisher's exact test two-tailed.).

DOI: https://doi.org/10.7554/eLife.34467.012

The following figure supplement is available for figure 5:

**Figure supplement 1.** Human scoring of PBEs and session quality.

DOI: https://doi.org/10.7554/eLife.34467.013

model-congruence approaches were varied (*Figure 5b*). A perfect detector would have an AUC of unity. We did not find a significant difference between the AUC of Bayesian decoding and model-congruence ($p = 0.14$, bootstrap, see Materials and methods). If we select thresholds such that our algorithms yield a similar fraction of significant vs. total events as the 24% denoted by our human scorers, we find that both Bayesian and model-congruence yield agreement of ≈70% labeled events with each other and with human scorers (*Figure 5c*).

Thus, congruence with an HMM trained only on PBEs appears to work as reliably as Bayesian decoding in detecting sequential reactivation of linear track behaviors. However, when we examined individual sessions, we noticed that performance was quite variable. Given that our models are learned only from PBEs, we reasoned that the statistics or structure of the PBEs within each session might yield models which vary in quality depending on the number of recorded units, the number of PBEs detected, and their self-consistency across events. We created a model quality metric by comparing cross-validated learning statistics to models which were learned from shuffled events (see Materials and methods). We found that the performance of model-congruence detection was tied to model quality ($R^2 = 0.17$, $F = 2.9$, $n = 18$ sessions, *Figure 5—figure supplement 1*). Model quality, in turn, was highly correlated with the number of PBEs during the session ($R^2 = 0.96$, $F = 392.6$, $n = $

18 sessions, *Figure 5—figure supplement 1*). Not surprisingly, the performance of Bayesian decoding relative to human scorers was independent of the quality of the HMM, or the number of PBEs, as the place field model is learned from ensemble neural activity during running. Thus, we find an intriguing contrast—when there is an abundance of PBEs (indicating novelty, learning, hippocampus-dependent planning, etc. [*Buzsáki, 2015*]), even in the absence of repeated experience, replay detection based on PBE activity is highly effective. Conversely, when there are few PBEs (i.e., scenarios in which PBEs are uncorrelated with cognitive function), but an abundance of repeated behavioral trials, Bayesian decoding of these limited events proves more effective.

## Modeling internally generated activity during open field behavior

The linear track environment represents a highly-constrained behavior. We therefore asked whether the HMM approach could generalize to more complex environments and behavioral tasks. (*Pfeiffer and Foster, 2013*, *Pfeiffer and Foster, 2015*) had previously recorded activity of CA1 neurons in rats as they explored in a 2 m $\times$2 m open field arena for liquid reward. Briefly, animals were trained to discover which one of 36 liquid reward wells would be the 'home' well on a given day. Then, they then were required to alternate between searching for a randomly rewarded well and returning to the home well. Using the place cell map in this task and Bayesian decoding, many PBEs were decoded to trajectories through two-dimensional space that were predictive of behavior and shaped by reward. Using this same dataset, we trained a HMMs on neural activity during PBEs in the open field. Here, we used the same PBE detected previously (*Pfeiffer and Foster, 2013*; *Pfeiffer and Foster, 2015*) which occupied an average of $2.53 \pm 0.42\%$ of the periods during which animals were behaving ($77.91 \pm 21.16$ s out of $3064.86 \pm 540.26$ s). Given the large number of units available in this dataset and the increased behavioral variability in the open field environment compared to the linear track, we chose to estimate HMMs with 50 latent states. The transition matrix and observation model from a sample session are shown in *Figure 6a,b*. Despite the complex and varied trajectories displayed by animals, the HMM captured sequential dynamics in PBE activity, as in the 1D case, when we compared learned models against both actual and surrogate test data, we found that the model likelihood was significantly greater for real data ($p<0.001$, Wilcoxon signed-rank test).

In the case of the linear track, we linked sparsity of the transition matrix to the sequential nature of behaviors in that environment. An unconstrained, two-dimensional environment permits a much richer repertoire of behavioral trajectories. However, behavior is still constrained by the structure of space—arbitrary teleportation from one location to another is impossible. We found that learning from PBEs in the open field yielded transition matrices (*Figure 6a*) that were significantly sparser thanmodels learned from shuffled data ($p<0.05$, Wilcoxon signed-rank test, $n = 8$ sessions, *Figure 5—figure supplement 1*). However, consistent with increased freedom of potential behaviors, when we compared the sparsity of models learned from open field acpPBEs with 50-state models learned from PBEs in linear tracks, the open field transition matrices were less sparse ($p<0.001$, Mann–Whitney $U$ test comparing 8 and 18 sessions, *Figure 4—figure supplement 1*). Likewise, when we examined the observation model for the open field, we found that the activity across states for individual neurons was significantly more sparse than in models learned from shuffled data ($p<0.05$, Wilcoxon signed-rank test, $n = 8$ sessions, *Figure 6—figure supplement 1*). The sparsity of linear track and open field observation models were not significantly different ($p = 0.44$, Mann–Whitney $U$ test).

Do the latent states learned from PBEs capture spatial information in a 2D environment? We used the PBE-trained model to decode run data, as in the linear track case. We found that the latent states corresponded with specific locations in the open field, as we expected (*Figure 6c*). Moreover, we were able to decode animals' movements with significantly greater than chance accuracy by converting decoded latent states to positions using the lsPF ($p<0.001$, *Figure 6d*). Finally, we examined model-congruency for PBEs detected in the open field. Previously, it was reported that 27.3% (815 of 2980, $n = 8$ sessions) were identified as 'trajectory events' (*Pfeiffer and Foster, 2015*). We chose a significance threshold to match this fraction (*Figure 6—figure supplement 3*) and found that there was significant overlap between the events detected through Bayesian and model-congruence techniques ($p<0.01$, Fisher's exact test). These events overlapped significantly with replay events detected using traditional Bayesian decoding (*Figure 6—figure supplement 3*). Thus, an HMM of the activity during population bursts captures the structure of neural activity in two dimensional

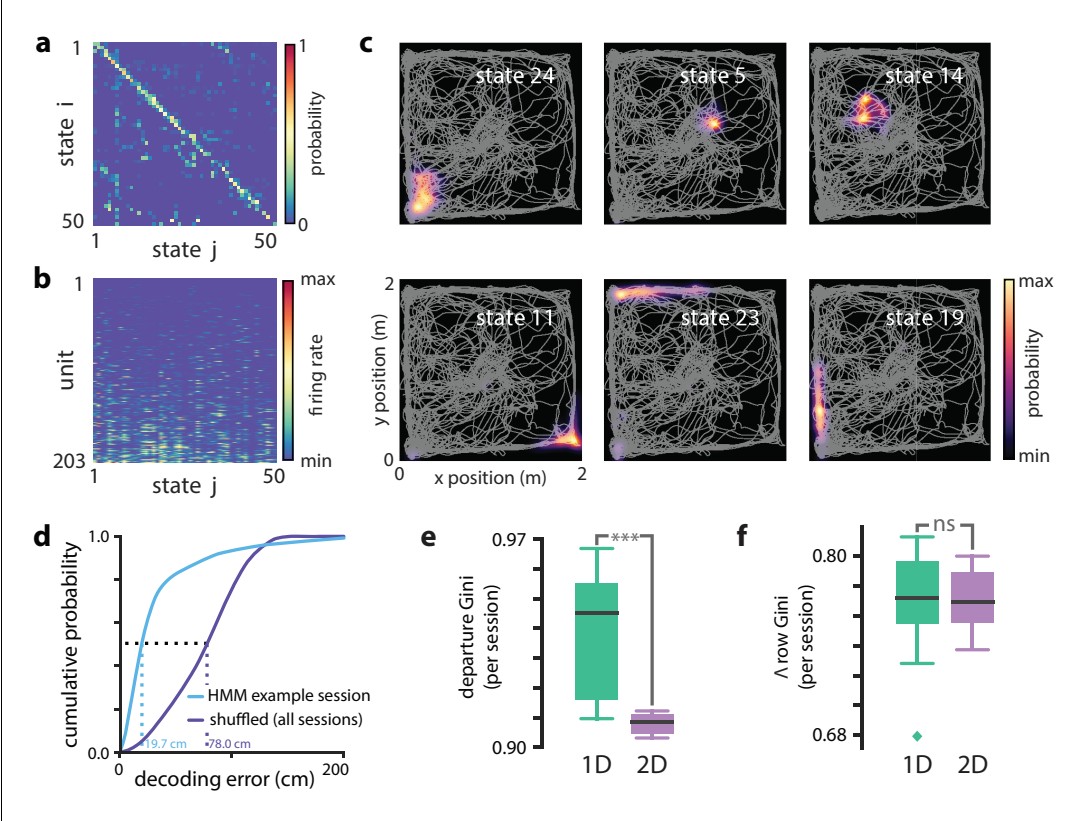

**Figure 6.** Modeling PBEs in open field. Modeling PBEs in open field. (**a**) The transition matrix estimated from activity detected during PBEs in an example session in the open field. (**b**) The corresponding observation model (203 neurons) shows sparsity similar to the linear track. (**c**) Example latent state place fields show spatially-limited elevated activity in two dimensions. (**d**) For an example session, position decoding through the latent space gives significantly better accuracy than decoding using the shuffled latent state place fields. (**e**) Comparing the sparsity of the transition matrices (mean Gini coefficient of the departure probabilities) between the linear track and open field reveals that, as expected, over the sessions we observed, the open field is significantly *less sparse* ($p<0.001$), since the environment is less constrained. (**f**) In contrast, there is not a significant difference between the sparsity of the observation model (mean Gini coefficient of the rows) between the linear track and the open field. Note that the linear track models are sparser than in *Figure 2* due to using 50 states rather than 30 to match the open field.

DOI: https://doi.org/10.7554/eLife.34467.014

The following figure supplements are available for figure 6:

**Figure supplement 1.** Open field PBE model states typically only transition to a few other states.
DOI: https://doi.org/10.7554/eLife.34467.015
**Figure supplement 2.** Each neuron is active in only a few model states in the open field.
DOI: https://doi.org/10.7554/eLife.34467.016
**Figure supplement 3.** lsPF and position decoding in an open field.
DOI: https://doi.org/10.7554/eLife.34467.017
**Figure supplement 4.** Examples of open field PBEs.
DOI: https://doi.org/10.7554/eLife.34467.018

environments during complex tasks and can be used to decode events consistent with trajectories through that environment.

## Extra-spatial information

As described earlier, while we observed a similar fraction of events to be similar by HMM-congruence and Bayesian decoding, there was not an exact event-to-event correspondence. An intriguing potential explanation is that the latent space represented in PBE sequential firing and captured by the HMM is richer than simply the spatial structure of the present environment. In most hippocampal ensemble recording experiments, maze or open field tasks are structured to intentionally map memory elements to spatial behavior, and thus this potential richness is difficult to test. We used two

sample datasets to explore the potential of the HMM to capture extra-spatial richness in the PBE sequences.

First, we considered the possibility that in the awake behaving animal, PBE activity might be sequential reactivation of environments other than the one being explored ('remote replay'). We reasoned that we could enhance the model's representation of remote environments by filtering out local replay from the training data. We evaluated how the model-quality of our HMM changed as progressively more sequences labeled as replay by Bayesian decoding were removed from the training data. In the linear track sessions we considered, we found that refining the training data resulted in models that lowered in quality at different rates as the threshold for Bayesian replay was decreased (*Figure 7*). Most, but not all, models dropped precipitously in quality: >50% when we removed events detected as Bayesian replay at a 95% threshold, as would be expected if the HMM represented only the local environment. In many outlier sessions in which model quality decreased more slowly, the initial (baseline)model quality was low. Intriguingly, however, in at least one outlier session where model quality decreased slowly with refinement (blue line,*Figure 7a*), the initial model quality was still high, and we further noted that position decoding using lsPF yielded relatively high error (blue dot, *Figure 7b*). Thus, we wondered whether this and similar sessions might have contained non-local or extra-spatial PBEs that were captured by the HMM.

In order to validate the concept of model-training refinement, we considered a dataset in which multiple environments were explored on the same day and remote replay was previously observed (*Karlsson et al., 2015*). These data consisted of a series of short exploratory sessions in which an animal first explored a novel maze (E1) and then was placed in a familiar one (E2). We identified awake PBEs during the familiar E2 session and used them to train an HMM. When we refined this model by removing Bayesian-significant local replay events from the training data, we found that the model quality decreased comparatively slowly (*Figure 7a*, green line), indicating that the HMM was capturing more than the local spatial structure. In contrast, when we used place fields from E1 to identify Bayesian-significant remote replay events and removed these from the training data, we found that the model quality decreased rapidly as with the general linear track cases (*Figure 7a*, red line). When we examined individual events in detail in this data, we found many examples in which HMM-significant, Bayesian non-significant PBEs decoded to extended state sequences which turned out to correspond to reactivation of the remote track (two are shown in *Figure 7c–l*). If we imagine that in this experiment data were only recorded during exploration of the familiar environment, classical Bayesian decoding would treat these events as noise, as shown in the bottom half of the two examples. In contrast, our HMM-based analysis finds these events to be significant, as shown in the top half of the two examples. Thus, by combining classical Bayesian decoding and HMM-congruence, we are able to identify a signature of when a HMM trained on PBEs captures sequential structure distinct from that dictated by the local environment. Additionally, in these cases, we show that specific non-local reactivation events can be identified.

Finally, we considered the potential of our methodology for uncovering temporal patterns in PBE activity under scenarios where complex behavior does not permit identification of well-defined place-fields or in the absence of behavior, such as during sleep. As we have emphasized, a remarkable aspect of learning HMMs from PBE activity is that the model can be built entirely without behavioral data, so can our model capture significant sequential information outside of immobility periods during quiet waking? To demonstrate this potential, we examined HMMs trained on PBEs in sleep following the learning phase of an object-location memory task when animals explored three objects in an open field (see Material and methods). Previous studies have demonstrated that subsequent recall of this memory is hippocampus-dependent, and requires consolidation in post-task sleep (*Prince et al., 2014*; *Inostroza et al., 2013*). However, while this task involves spatial exploration of objects in an arena, whether the subsequent post-task sleep contains sequential structure and whether object memory is contained in this code has remained elusive (*Larkin et al., 2014*). In order to assess the presence of sequential structure in the PBEs, we first used cross validation to generate a distribution of sequence HMM-congruence scores. For each set of test PBEs, we also generated surrogates by shuffling time bins across events (pooled time-swap). Using our HMM-congruence score which explicitly tests for sequences through state space, the large difference between actual and shuffled score distributions indicates evidence for significant sequential structure in the PBEs ($p<0.001$, Mann–Whitney $U$ test, *Figure 8*). While more work is needed to evaluate the mnemonic relevance of these HMM-congruent sequences, these data support the notion that the HMM

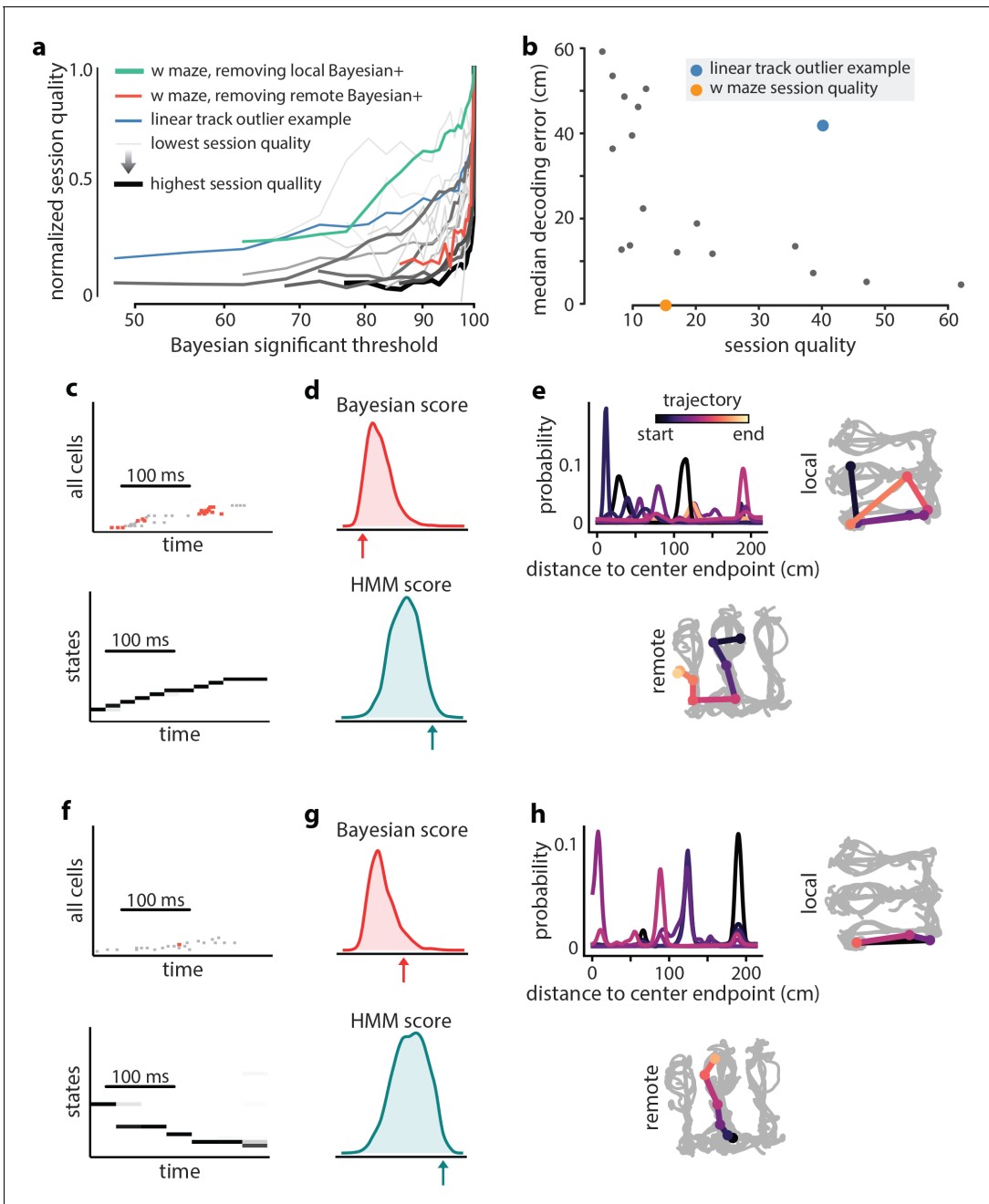

**Figure 7.** Extra-spatial structure. Examples of remote replay events identified with HMM-congruence. We trained and evaluated HMMs on the events that were not Bayesian significant (residual events) to identify potential extra-spatial structure. (**a**) The normalized session quality drops as local-replay events above the Bayesian significance threshold are removed from the data. Each trace corresponds to one of the 18 linear track sessions, with the stroke width and the stroke intensity proportional to the baseline (all-events) session quality. The blue line identifies a session in which model quality drops more slowly, indicating the potential presence of extra-spatial information. The reduction in session quality for a W maze experiment with known extra-spatial information is even slower (green). When, instead, Bayesian-significant *remote* events are removed, rapid reduction in session quality is again revealed (red). (**b**) The lsPF-based median decoding errors are shown as a function of baseline session quality for all 18 linear track sessions. The blue dot indicates the outlier session from panel (**a**) with potential extra-spatial information: this session shows high decoding error combined with high session quality. Session quality of the W maze session is also indicated on the x-axis (decoding error is not directly comparable). (**c–n**) Two example HMM-congruent but not Bayesian-significant events from the W maze session are depicted to highlight the fact that congruence can correspond to remote replay. (**c**) Spikes during ripple with local place cells highlighted (top panel) and the corresponding latent state probabilities (bottom panel) decoded using the HMM show sequential structure (grayscale intensity corresponds to probability). (**d**) In this event, the Bayesian score relative to the shuffle distribution (top panel) indicates that the event is not-significant, whereas the HMM score relative to shuffles indicates (bottom panel) the ripple event is HMM-congruent. (**e**) Estimates of position using local place fields show jumpy, multi-modal *a posteriori* distributions over space in 1D (top left

*Figure 7 continued on next page*

*Figure 7 continued*
panel) and 2D (top right panel; distribution modes and time is denoted in color). Bayesian decoding using the remote environment place fields (bottom panel) indicates that the sample event is a remote replay. Note that in a typical experiment, only the local place fields would be available. (**f–h**) Same as (**c–e**), but for a different ripple event.
DOI: https://doi.org/10.7554/eLife.34467.019

can uncover sequential activity in sleep away from the task environment. This approach further demonstrates the utility of the HMM approach as an initial analysis of a novel dataset, or as a way of comparing the sequential content encoded in PBEs during different periods.

## Discussion

Increasing lines of evidence point to the importance of hippocampal ensemble activity during PBEs in guiding on-going behavior and active learning. Despite being the strongest output patterns of the hippocampus, however, this activity has been assumed to be interpretable only in the context of other theta-associated place cell activity expressed during behavior. Our findings demonstrate that over the course of a behavioral session, ensemble activity during PBEs alone is sufficient to form a model which captures the spatial relationships within an environment. This suggests that areas downstream of the hippocampus might be able to make use solely of PBE activity to form models of external space. In an extreme view, place cell activity might merely subserve the internal mechanisms in the hippocampus which generate PBE sequences. To the extent that animals might wish to use the spatial code obtained from PBEs to identify their current location, we show that this can be done after translating ensemble activity into the latent states of the model. Do the PBEs contain 'full information' about the environment? Bayesian decoding of location from place cell activity results in lower error than location estimates generated using the latent states and lsPF. This suggests that the manifold defined by the HMM may not capture all the dimensions of information represented during exploration. However, it is possible that with more PBE data, we would learn a more refined state space. Thus, the difference between the latent space represented during behavior and within PBEs may be an interesting focus of future study.

When we examined the transition matrices we learned from PBEs, we found that they were marked by significant sparsity. This sparsity results from the sequential patterns generated during PBEs. Latent variable models have previously been used to analyze the structure of hippocampal place cell activity (*Chen et al., 2012*; *2014*; *Dabaghian et al., 2014*). In these studies, the learned transition matrices were mapped to undirected graphs which could be analyzed using topological measures. It is intriguing that similar structure is apparent in PBE activity. For example, we observed that transition matrices learned from PBEs associated with linear track behavior were significantly sparser than those learned from the open field, which we hypothesize is a consequence of the greater freedom of behavior in the latter (a topological difference). Whether hippocampal PBE activity must always be sequential, i.e., evolve through a sparsely-connected latent space, is an open and interesting question, as are differences between the latent state space dynamics learned during PBEs and those learned from place cell activity.

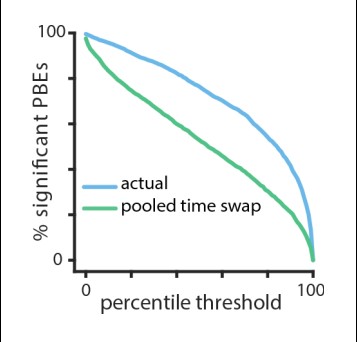

**Figure 8.** Temporal structure during a sleep period following object-location memory task. Using cross validation, we calculate the HMM-congruence score (which ranges from 0 to 1) for test PBEs. For each event, we also calculate the score of a surrogate chosen using a pooled time-swap shuffle across all test events. The distribution of scores of actual events is significantly higher than that of the surrogate data (*p*<0.001, Mann–Whitney *U* test).
DOI: https://doi.org/10.7554/eLife.34467.020

## Graded, non-binary replay detection

Remarkably, evaluating the congruence or likelihood of test data against our HMM provided a highly novel method to detect events that are consistent with replay, without a need to access the 'play' itself. In the process of evaluating the potential of HMMs for detecting replay, we developed an approach to compare different replay-detection strategies. Our results highlight how the data do not readily admit to a strict separation between 'replay' and 'non-replay' events. While it is possible that with additional shuffles or other restrictions (*Silva et al., 2015*), automated performance might be rendered closer to human-labeling, even human scorers had variation in their opinions. This calls into doubt judgments of memory-related functions which build on a binary distinction between replay and non-replay sequences. Model congruence, either as a raw statistical likelihood or weighted against a shuffle distribution, seems to be a very reasonable metric to associate with individual PBEs. Moreover, evaluating congruence with an HMM does not require access to repeated behavioral sequences, which may be infeasible under widely-used single- or few-trial learning paradigms or when the events involve replay of a remote internalized environment. Given these benefits, along with computational efficiency, we would suggest that future analyses of the downstream impact of hippocampal reactivation regress effects against this measure rather than assuming a binary distinction.

## Learning, Model Congruence and Replay Quality

Not surprisingly, the rate of PBEs had a large effect on our ability to measure model congruence. Interestingly, it has been noted that the density of PBE is higher during early exposure to a novel environment (*Cheng and Frank, 2011*; *Frank et al., 2004*; *Kemere et al., 2013*; *Kudrimoti et al., 1999*). This might suggest that for the animal, PBE activity could be an important source for generating models of the world when the animal is actively learning about the environment. If as hypothesized, replay is a form of rehearsal signal generated by the hippocampus to train neocortical modules (*McClelland et al., 1995*; *Buzsáki, 1989*), then indeed the brain's internal machinery may also be evaluating whether a given sequential PBE pattern is congruent and consistent with previously observed PBEs. In later sessions, as animals have been repeatedly exposed to the same environments, downstream regions will have already witnessed many PBEs from which to estimate the structure of the world. Overall, our approach provides a novel viewpoint from the perspective of hippocampal PBEs. An interesting future line of inquiry would be to assess the extent to which a model built on PBEs during first experience of a novel environment is slower or faster to converge to the final spatial map than models built on theta-associated place activity.

## Application to Extra-spatial behaviors

We have analyzed data gathered in experiments in which rats carried out simple spatial navigation tasks. Thus, to some extent it is not surprising that when we decoded ensemble activity during behavior we found that spatial positions the animal is exploring are strongly associated with the latent states.

We anticipate that our approach for calculating lsPF would be equally useful in tasks in which the hippocampal map is organized around time (*Eichenbaum, 2014*; *Rodriguez and Levy, 2001*) or other continuous variables (e.g. sound frequency [*Aronov et al., 2017*]). Our two proof-of-concept analyses, however, suggest that it should be possible to use HMMs to infer the presence of extra-spatial sequential reactivation in PBEs. For example, we showed that there is significant sequential structure during sleep after an animal explores novel objects in an environment. We anticipate that careful experimental design and further algorithmic development would allow for the conjunctive coding of object identity and spatial locations to be detected in the latent states we learn from PBEs, with model-congruence providing a tool to study sequential hippocampal reactivation in these types of tasks.

Conjunctive, non-spatial information might be one source of the apparent variability that results in many PBEs not being detected as replay using traditional Bayesian decoding. Another proposed source of this variability is reactivation of other environments. Our second proof-of-concept analysis suggests that HMMs learned from PBEs can, in fact, capture the spatial structure of environments beyond the one the animal is currently exploring. It appears that it should be possible to use only the PBEs and information about the place-cell map of the local environment to refine the training set

for remote replay activity and learn the structure of a remote environment. While we used Bayesian decoding to detect putative local replays, we anticipate related approaches might use an HMM or other approaches to model local place cell activity.

### Future possibilities

It has been previously observed that the rate of hippocampal reactivations in PBEs during awake behavior is much higher than during sleep (*Grosmark and Buzsáki, 2016*; *Karlsson and Frank, 2008*), but the reasons for this are not well understood. One hypothesis is that many sleep PBEs contain the reactivation of contexts other than those measured during a behavioral experiment. Another hypothesis is that sleep activity involves remodeling of dynamic network architectures (*Buhry et al., 2011*; *Tononi and Cirelli, 2014*). Our approach has the potential to illuminate some sources of variability during sleep. While we have given preliminary evidence that information about a remote context can be present in PBEs along with the local context, further work is required to understand how our model's ability to capture this structure scales with the number of different contexts. With sufficient data, our HMM approach should be able to learn disjoint sets of latent states (or 'sub-models') which would capture these separate contexts and allow us to test this possibility. Alternatively, sleep PBEs could yield models which represent a known behavioral context but are markedly different (e. g., less sparse) than those learned from awake PBEs. This might support the network remodeling function of sleep. In the latter case, we might imagine that only a small subset of sleep PBEs—corresponding to learning-related replay—would be congruent with a model learned from awake PBE data.

### Conclusions

We have demonstrated a new analytical framework for studying hippocampal ensemble activity which enables primacy of PBEs in model formation. We use an unsupervised learning technique commonly used in the machine learning field to study sequential patterns, the hidden Markov model. This contrasts with existing approaches in which the model—estimated place fields for the ensemble—is formed using the theta-associated place cell activity. We find that our PBE-first approach results in a model which still captures the spatial structure of the behavioral tasks we studied. Additionally, we demonstrate that we can use model-congruence as a tool for assessing whether individual PBEs contain hippocampal replay. Finally, we present proofs-of-concept that this analytical approach can detect the presence of sequential reactivation in experimental scenarios in which existing approaches are insufficient. Thus, the use of unsupervised learning of latent variable models—specifically HMMs and statistical congruence as a marker of individual events—bears much promise for expanding our ability to understand how PBEs enable the cognitive functions of the hippocampus.

## Materials and methods

### Experiment paradigm/Neural data recording

We analyzed neural activity recorded from the hippocampus of rats during periods in which they performed behavioral tasks in different paradigms. First, we considered data from animals running back and forth in a linear track 150 or 200 cm long. As previously reported using these same data (*Diba and Buzsáki, 2007*), we recorded neural activity using chronically-implanted silicon probes to acquire the activity of hippocampal CA1/CA3 neurons. From these experiments, we chose sessions during which we observed at least 20 place cells during active place-field exploration, and at least 30 PBEs (see below). Place cells were identified as pyramidal cells which had (i) a minimum peak firing rate of 2 Hz, (ii) a maximum mean firing rate of 5 Hz, and (iii) a peak-to-mean firing rate ratio of at least 3, all estimated exclusively during periods of run (as defined before, that is, when the animal was running >10 cm/s). This selection yielded $n = 18$ session with 41–203 neurons (36–186 pyramidal cells). All procedures were approved by the Institutional Animal Care and Use Committee of Rutgers University and followed US National Institutes of Health animal use guidelines (protocol 90–042).

A second dataset used tetrodes to record a large number (101–242) of putative pyramidal neurons in area CA1 during two sessions each in four rats. Briefly, as was previously reported using these data (*Pfeiffer and Foster, 2013*, *Pfeiffer and Foster, 2015*), rats explored an arena in which

there were 36 reward sites. In each session, one site was designated as 'home'. During a session, rats would repeatedly alternate between retrieving a random reward site in one of the remaining 35 locations and retrieving a reward at the home location. All procedures were approved by the Johns Hopkins University Animal Care and Use Committee and followed US National Institutes of Health animal use guidelines (protocols RA08M138, RA11M16, and RA14M48).

In order to investigate remote replay, we used data from an experiment in which this phenomenon has been previously reported (*Karlsson and Frank, 2009*). Briefly, rats were implanted with multi-electrode microdrives with tetrodes targeting CA1 and CA3. They were trained to carry out a continuous-alternation task in an initially novel 'w'-shaped maze (E2) for liquid reward for multiple daily run sessions interspersed by rest-periods in an enclosed box. After they learned the task, they were introduced to a novel w-maze (E1) in a different orientation in which they had two run sessions followed by a run in the now-familiar E2. For our proof-of-concept analysis (*Figure 7*), we used data from the second day of the novel maze (i.e., third and fourth exposures) in animal 'Bon'.

Finally, we recorded neural activity during an object-location memory task using a 32-channel silicon probes (Buzsaki32, Neuronexus, MI) equipped with light fibers lowered to area CA1 of the dorsal hippocampus. The animal was previously infused with AAV-CamKIIa-ArchT-GFP for the purpose of another experiment. Putative pyramidal cells and interneurons were distinguished based on their spike waveforms and spike auto-correlograms. On the day before the recordings, the animal was repeatedly exposed to an empty test chamber on four successive six minute blocks, interleaved by three minute rest periods in the home cage. On the recording day, the first of these six-minute blocks was again the empty test chamber, but on the remaining blocks, the animal was exposed to a fixed configuration of three different novel objects placed in the northeast, center and southeast corners of the box. These blocks were again interleaved with three minute rest periods in the home cage. The test chamber was a $60 \times 60$ cm$^2$ box with a local cue (8.5 in. $\times$ 11 in. sheet printout) placed on one test wall. Following the last acquisition exposure, the animal was returned to its home cage for a four hour extended sleep period. The subsequent day, one of the objects in the box was displaced and the animal was reintroduced into the box to test for interactions with the displaced versus non-displaced objects. All procedures were approved by the Institutional Animal Care and Use Committee of the University of Wisconsin-Milwaukee and followed US National Institutes of Health animal use guidelines (protocol 13–14 #28)

## Population burst events

To identify PBEs in the linear track data, a SDF was calculated by counting the total number of spikes across all recorded single and multi-units in non-overlapping 1 ms time bins. The SDF was then smoothed using a Gaussian kernel (20 ms standard deviation, 60 ms half-width). Candidate events were identified as time windows with a peak SDF of at least three standard deviations above the mean calculated over all the session. The boundaries of each event were set to time points of crossing the mean, preceding and following the peak. Events during which animals were moving (average movement speed of >5 cm/s) were excluded from all further analyses to prevent possible theta sequences from biasing our results. For analysis, we then binned each PBE into 20 ms (non-sliding) time bins. Spikes from putative interneurons (mean firing rate when moving >10 Hz) were excluded, as were events with duration less than four time bins or with fewer than four active pyramidal cells. For the open field data, we used the previously reported criteria (*Pfeiffer and Foster, 2013*) for identifying PBEs prior to binning (10 ms standard deviation kernel, minimum of 10% of units active, duration between 50 ms and 2000 ms).

## Hidden markov model of PBE activity

We trained HMMshe complete sequence on the PBEs. In an HMM, an unobserved discrete latent state $q_t$ evolves through time according to a first order Markov process. The temporal evolution of the latent state is described by the $M \times M$ matrix $\mathbf{A}$, whose elements $\{a_{ij}\}$ signify the probability after each time bin of transitioning from state $i$ to state $j$, $a_{ij} = \Pr(q_{t+1} = j | q_t = i)$. The number of states, $M$, is a specified hyperparameter. We found that our results were insensitive to the value of $M$ through a wide range of values from 20 to 100 (*Figure 3—figure supplement 1*). During each time bin of an event, the identity of the latent state influences what is observed via a state-dependent probability distribution. We modeled the $N$-dimensional vector of binned spiking from our

ensemble of $N$ neurons at time $t$, $O_t$, as a Poisson process. Specifically, for each state, $i$, we model neuron $n$ as independently firing according to a Poisson process with rate $\lambda_{ni}$.

$$\Pr(O_t|q_t = i) = \prod_{n=1}^{N} \Pr(o_{nt}|q_t = i) \propto \prod_{n=1}^{N} (\lambda_{ni})^{o_n} exp(-\lambda_{ni})$$

where $o_{nt}$ is the number of spikes observed from neuron $n$ at time $t$. The final parameter which specifies our model is the probability. distribution of the initial state for a given event, $\pi_i = Pr(q_1 = i)$. Thus, our model is specified by parameters $\theta = \{\mathbf{A}, \Lambda, \pi\}$, where $\Lambda = \{\lambda_{ni}\}$ is an $N \times M$ matrix and $\pi = \{\pi_i\}$ is an $N$-dimensional vector.

To learn model parameters, we follow the well-known iterative EM procedure (*Rabiner, 1989*), treating each training PBE as an observation sequence. In order to regularize the model, we impose a minimum firing rate for each neuron of 0.001 (0.05 Hz) during the M-step of EM. For a given PBE (i.e., observation sequence) with $K$ bins, we use the 'forward-backward algorithm' (*Rabiner, 1989*) to calculate the probability distribution of the latent state for each time bin, $\Pr(q_t|O_1, \ldots, O_t, \ldots, O_K)$. For a particular time bin, $t$, in a given sequence, the forward-backward algorithm allows information from all observation bins, previous and subsequent, to affect this state probability distribution (as well the observation bin at time $t$). The forward-backward algorithm also efficiently calculates the 'score', or likelihood of the complete sequence, $\Pr(O_1, \ldots, O_K)$. All HMMs learned in this work used five-fold cross validation, that is, the PBEs were divided into five randomly selected fifths ('folds'), and then each fold was evaluated as a test set, with the model trained using the remaining four folds. We define the model likelihood of an HMM as the product of the scores of each event using this five-fold cross validation. To initially evaluate model learning, we compared model likelihoods calculated using real and shuffled test data. Models which have learned to properly represent the data should show significant increases. To quantify the presence of PBE sequences in a model we used a model quality metric as described below.

## Ordering states for visualization

For visualization, we wanted to order the states to maximize the super diagonal of the transition matrix. We used a greedy approach which typically yields this solution. We started by assigning the first index to the state with the highest initial probability and added states based on the most probable state transitions. The undirected connectivity graphs were then generated from this transition matrix, averaging the strength of reciprocal connections, $a_{ij}$ and $a_{ji}$.

## Surrogate datasets and shuffle methods

In order to analyze the HMMs we learned, we compared them against different types of surrogate datasets obtained by shuffling the neural activity during PBEs in distinct ways. (1) Temporal shuffle: within each event, the binned spiking activity was circularly permuted across time for each unit, independently of other units. This goal of this shuffle is to disrupt unit co-activation, while maintaining the temporal dynamics for each unit. (2) Time-swap shuffle: within each event, the order of the binned columns of neural activity was randomly permuted across time, coherently across units. The goal of this shuffle is to change the temporal dynamics of ensemble activity, while maintaining unit co-activation. (3) Poisson surrogate 'shuffle': we estimated each unit's mean firing rate across all PBEs, and then produced surrogate PBEs from independent Poisson simulations according to each unit's mean firing rate. (4) Pooled time-swap shuffle: the order of the binned columns of neural activity was randomly permuted across all pooled events, coherently across units. This shuffle has been previously used in Bayesian replay detection (*Davidson et al., 2009*).

## Calculating sparsity and connectivity of the model parameters

Sparsity of the transitions from individual states (departure sparsity) was measured by calculating the Gini coefficient of corresponding rows of the transition matrix (*Hurley and Rickard, 2009*). The Gini coefficient is a measure of how variable the values of this probability distribution are, with equality across states corresponding to a coefficient of zero (minimal sparsity), and a singular distribution with a probability-one transition to a single other state corresponding to a coefficient of one (maximal sparsity). The sparsity of the full transition matrix was calculated by averaging the Gini coefficient across rows. For analyses of PBE models from linear tracks, we computed the mean sparsity across

states for each of the 250 surrogate datasets, and these means were used to generate the box plots of *Figure 2c*. Note that for the actual data, we generate a distribution by randomly initializing the model 250 times and calculating the mean sparsity over all initializations. For analyses of models learned from PBEs in open fields (and the linear track comparison with 50 states), we created 50 surrogates/random initializations (*Figure 6—figure supplement 1*). To compare across sessions, we calculated the mean sparsity by averaging over all 250 surrogate datasets to obtain a single mean sparsity per session, so that $n = 18$ per-session means were used to create the box-plots of *Figure 2e*.

Firing rates can be highly variable for different units. Thus, when evaluating the sparsity of the observation matrix, we measured the extent to which individual units were specifically active in a few states by calculating the Gini coefficients of the rows of the observation matrix. As with transitions, we calculated mean sparsity across units for each surrogate dataset (e.g., linear track, *Figure 2d*; open field, *Figure 6—figure supplement 2*), and we then averaged over all surrogate datasets to obtain aper-session average, used in *Figure 2f*.

## Model connectivity and sequences

To measure the degree of sequential connectivity within the graph corresponding to the transition matrix—with nodes and edges representing the states and transitions, respectively—we developed an algorithm for measuring the length of the longest path that can be taken through the graph. This method is analogous to the 'depth-first search' algorithm for traversing the graph's tree structure without backtracking. First, we made an adjacency matrix for a corresponding unweighted directed graph by binarizing the transition matrix using a threshold of 0.2 on the transition probabilities. Starting from each node, we then found the longest path that ended at either a previously visited node or a terminal node (a node without any outgoing edges). To compare models trained on actual versus surrogate datasets, we adjusted the thresholds to match the average degree (defined as the average number of edges per node) between the models, thus ruling out possible effects due to differences in the number of graph edges. We carried out this analysis on the same set of models that were generated for analyzing sparsity. To compare across sessions, we calculated the median maximum path length for each session ($n = 18$) and used the per-session medians to generate box plots of *Figure 2—figure supplement 3c*.

## Latent state place fields

To calculate the latent state place fields, we first identified bouts of running by identifying periods when animals were running (speed >10 cm/s). We then binned the spiking during each of these bouts in 100 ms bins. Using the forward-backward algorithm (*Rabiner, 1989*) and the HMM model parameters learned from PBEs, we decoded each bout into a sequence of latent state probability distributions, $\Pr(q_t|O_t)$. Using the track positions corresponding to each time bin, we then found the average state distribution for each position bin, $x_p$, and normalized to yield a distribution for each state, $\Pr(x_p|q_t = i)$.

## Decoding position from latent state sequences

We used the lsPF to decode the animal's position after determining the probability of different latent state trajectories during bouts of running. With five-fold cross validation, we estimated lsPF in a training dataset, then used the HMM model to decode latent state trajectory distributions from ensemble neural activity in the test data. The product of lsPF and decoded latent state distribution at time $t$ is the joint distribution $\Pr(x_p, q_t|O_t)$. We decode position as the mean of the marginal distribution $\Pr(x_p|O_t)$.

## Bayesian replay detection

We followed a frequently used Bayesian decoding approach to detect replay in our 1D data (*Kloosterman, 2012*). For each 20 ms time bin $t$ within a PBE, given a vector comprised of spike counts from $N$ units, $O_t = (o_{1t}o_{2t}\ldots o_{Nt})$ in that bin, the posterior probability distribution over the binned track positions was calculated using Bayes' rule:

$$Pr(x_p|O_t) = \frac{Pr(O_t|x_p)Pr(x_p)}{\sum_{q=1}^{P} Pr(O_t|x_q)Pr(x_q)},$$

where $x_p$ is the center of $p$-th linearized position bin (of $P$ total bins). We assumed Poisson firing statistics, thus the prior probability, $\mathrm{Pr}(O_t|x_p,)$, for the firing of each unit $n$ is equal to

$$Pr(O_t|x_p) = \prod_{n=1}^{N} \mathrm{Pr}(o_{nt}|x_p,) \propto \prod_{n=1}^{N} (\tau\lambda_{np})^{o_{nt}} exp(-\tau\lambda_{np})$$

where $\tau$ is the duration of time bin (100 ms during estimation, 20 ms during decoding), and $\lambda_{np}$ characterizes the mean firing rate of the $n$-th unit in the $p$-th position bin. We assumed a uniform prior distribution $Pr(x_p)$ over the position bins.

For each PBE, the estimated posterior probability distribution was used to detect replay as follows. Many (35,000) lines with different slopes and intercepts were sampled randomly following the approach in (*Kloosterman, 2012*). The Bayesian replay score for a given event was the maximum score obtained from all candidate lines, where the score for a particular line was defined as the mean probability mass under the line, within a bandwidth (of 3 cm). For time bins during which the sampled line fell outside of the extent of the track, the median probability mass of the corresponding time bin was used, and for time bins during which no spikes were observed, we used the median probability mass across all on-track time bins. To evaluate the significance of this score, for each event we generated 5000 surrogates of the posterior probability distribution by cycling the columns (i.e., for each time bin, circularly permuting the distribution over positions by a random amount) and calculated the replay score for each surrogate. The Monte Carlo $p$-value for each event was obtained from the number of shuffled events with replay scores higher than the raw data. The threshold for significance was varied as described in the text. For the open field, we used previously reported criteria (*Pfeiffer and Foster, 2013*) to identify replay events from PBEs.

## Replay detection via PBE model congruence

To identify replay as model congruence, for each PBEs, we used the forward-backward algorithm to calculate the sequence likelihood $\mathrm{Pr}(O_1, \ldots, O_K)$, as defined earlier. Using five-fold cross validation, the parameters of a HMM were learned from training PBE. The sequence score was then calculated for each event in the test data. To evaluate the significance of this score, for each event we generated 5000 surrogate scores using a computationally-efficient scheme. Specifically, for each surrogate, we randomly shuffle the rows of the transition matrix, excepting the diagonal. By maintaining the diagonal (i.e., transitions that begin and end in the same state) and leaving the observation model unchanged, this shuffle specifically selects against PBEs in which the latent states do not evolve in temporal sequences. The Monte Carlo $p$-value for each event was calculated as the fraction of shuffled events with HMM sequence scores higher than the raw data. The threshold for significance was varied as described in the text. Note that while we describe this as HMM-congruence, we have maintained the diagonal of the transition matrix, which specifically selects against PBEs which might be model-congruent by maintaining a single state over many time bins. In reality there are other dimensions of the HMM that we could assess congruence against, for example the observation model, the initial state distribution, or combinations of these and the transition matrix. In comparing against Bayesian decoding, our current definition seemed most appropriate for sequence detection, but we can imagine future studies expanding on our approach.

## Human scoring and detection comparison

We organized a group of human scorers to visually evaluate whether individual PBEs should be described as replay. More specifically, scorers were only presented with Bayesian decoded probability distributions such as those in *Figure 4a*, but without access to the spike raster or any additional information. The scorers included six graduate students (including one of the authors) and two undergraduates, all of whom were generally familiar with the concept of hippocampal replay. We built an automatic presentation system which would display each event in random order, and record one of six possible scores: 'excellent' (highly sequential with no jumps and covering most of the track), 'good' (highly sequential with few or no jumps), 'flat' (decoded position stayed mostly in the

same place, i.e. no temporal dynamics), 'uncertain' (some semblance of structure, but not enough to fall into any of the previous categories) or 'noise' (no apparent structure, or nonsensical trajectories such as teleportation). An event was then designated as replay if it was labeled as 'excellent' or 'good' by a majority of scorers (ties were labeled as non-replay).

To calculate an ROC curve for replay detection algorithms, we used our shuffle statistics for each event to create a vector which related the significance threshold (e.g., 99%) to the label supplied by the algorithm (i.e., significant replay or not). Then, as a function of threshold, the sensitivity (fraction of true positives identified) and selectivity (fraction of true negatives identified) were averaged over events to yield an ROC curve. To evaluate whether the AUC differed between Bayesian and model-congruence techniques we used a bootstrap approach. To generate a null hypothesis, we combined the event/threshold vectors from both groups, and then sampled two random groups (A and B) with replacement from the pooled data. The AUC for these two random groups of events were measured, and a distribution for the difference between the randomly chosen AUC was calculated. The two-sided $p$-value we report is the fraction of differences in random AUC which are more extreme than the actual difference.

## HMM model quality across sessions

In order to understand the extent to which an HMM trained on PBEs from a given session contained sequentially-structured temporal dynamics, we calculated the 'session quality' (equivalently model quality) as follows. Again using five-fold cross validation, we learn an HMM on the training subset of PBEs, and score (using the forward-backward algorithm, as before), the remaining subset of test PBEs. Then, we also score a pooled time-swap surrogate of the test PBE and we repeat this pooled time-swap scoring $n = 2500$ times. Finally, we obtain a z score for each PBE by comparing the score from the actual test PBE, to the distribution of pooled time-swap scores of the corresponding PBE. The session quality is then defined as the average of these z scores, over all events in a session. This measure of session quality was then used to detect the presence of putative remote replay events or other extra-spatial structure in PBEs, since a high session quality after removing local Bayesian significant events is highly suggestive of remaining (unexplained) sequential structure.

## Software and data analysis

Data analyses were performed using MATLABr and Python. Jupyter notebooks (using Python) are available at https://github.com/kemerelab/UncoveringTemporalStructureHippocampus (copy archived at https://github.com/elifesciences-publications/UncoveringTemporalStructureHippocampus), where most of the results presented here are reproduced. We have also developed and open-sourced a Python package (namely nelpy) to support the analyses of electrophysiology data with HMMs, which is available from https://github.com/nelpy (*Ackermann et al., 2018*; copies archived at https://github.com/elifesciences-publications/nelpy, https://github.com/elifesciences-publications/tutorials, https://github.com/elifesciences-publications/example-analyses, https://github.com/elifesciences-publications/example-data and https://github.com/elifesciences-publications/test-data).

## Additional information

### Funding

| Funder | Grant reference number | Author |
| --- | --- | --- |
| National Science Foundation | IOS-1550994 | Etienne Ackermann<br>Caleb Kemere |
| Human Frontier Science Program | RGY0088 | Etienne Ackermann<br>Caleb Kemere |
| Ken Kennedy Institute | ERIT | Caleb Kemere |
| National Institute of Mental Health | R01MH109170 | Kourosh Maboudi<br>Kamran Diba |
| National Institute of Mental Health | R01MH085823 | Brad E Pfeiffer<br>David Foster |

| Alfred P. Sloan Foundation | | Brad E Pfeiffer<br>David Foster |
| --- | --- | --- |
| Brain and Behavior Research Foundation | NARSAD Young Investigator Grant | Brad E Pfeiffer<br>David Foster |
| McKnight Endowment Fund for Neuroscience | | David Foster |
| National Science Foundation | CBET-1351692 | Etienne Ackermann |

The funders had no role in study design, data collection and interpretation, or the decision to submit the work for publication.

### Author contributions
Kourosh Maboudi, Conceptualization, Data curation, Software, Methodology, Writing—original draft, Writing—review and editing; Etienne Ackermann, Conceptualization, Resources, Data curation, Software, Visualization, Methodology, Writing—original draft, Writing—review and editing; Laurel Watkins de Jong, Brad E Pfeiffer, Gathered experimental data, Data curation, Data curation, Writing—review and editing; David Foster, Data curation, Writing—review and editing; Kamran Diba, Conceptualization, Gathered experimental data, Data curation, Supervision, Methodology, Writing—original draft, Writing—review and editing; Caleb Kemere, Conceptualization, Software, Supervision, Methodology, Writing—original draft, Writing—review and editing

### Author ORCIDs
Kourosh Maboudi ID http://orcid.org/0000-0001-9675-4031
Kamran Diba ID https://orcid.org/0000-0001-5128-4478
Caleb Kemere ID http://orcid.org/0000-0003-2054-0234

### Ethics
Animal experimentation: As reported previously, all procedures were approved by the Johns Hopkins University, Rutgers University, and University of California, San Francisco Animal Care and Use Committees and followed US National Institutes of Health animal use guidelines (protocol 90-042).

### Decision letter and Author response
Decision letter https://doi.org/10.7554/eLife.34467.025
Author response https://doi.org/10.7554/eLife.34467.026

## Additional files

### Supplementary files
• Transparent reporting form
DOI: https://doi.org/10.7554/eLife.34467.021

### Data availability
We analyzed data from neural recording experiments. Data for Figures 1-5 has been previously reported in Diba and Buzsáki (2007, Nature Neuroscience). These data and also data for Figure 8 are available from Kamran Diba on request. Data for Figure 6 was previously reported in Pfeiffer and Foster (2015, Science) and is available from Brad Pfeiffer and David Foster on request. Data for Figure 7 was previously reported in Karlsson and Frank (2008, Nature Neuroscience) is available from the CRCNS.org archive ('hc-6'). All analysis code and sample recording epochs for Figures 1-7 are available on https://github.com/kemerelab/UncoveringTemporalStructureHippocampus (copy archived at https://github.com/elifesciences-publications/UncoveringTemporalStructureHippocampus). These make use of our broader open-source Python analysis software https://github.com/nelpy (copies archived at https://github.com/elifesciences-publications/nelpy, https://github.com/elifesciences-publications/tutorials, https://github.com/elifesciences-publications/example-analyses, https://github.com/elifesciences-publications/example-data, https://github.com/elifesciences-publications/test-data).

The following previously published dataset was used:

| Author(s) | Year | Dataset title | Dataset URL | Database, license, and accessibility information |
|-----------|------|---------------|-------------|--------------------------------------------------|
| Karlsson M, Carr M, Frank LM | 2015 | Simultaneous extracellular recordings from hippocampal areas CA1 and CA3 (or MEC and CA1) from rats performing an alternation task in two W-shaped tracks that are geometrically identically but visually distinct. | http://dx.doi.org/10.6080/K0NK3BZJ | Publicly available at CRCNS - Collaborative Research in Computational Neuroscience |

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
