## [Decision Letter]

[Editors' note: the authors’ plan for revisions was approved and the authors made a formal revised submission.]

Thank you for sending your article entitled "Uncovering temporal structure in hippocampal output patterns" for peer review at *eLife*. Your article is being evaluated by three peer reviewers, one of whom is a member of our Board of Reviewing Editors and the evaluation is being overseen by a Reviewing Editor and Michael Frank as the Senior Editor.

Given the list of essential revisions, including new experiments, the editors and reviewers invite you to respond within the next two weeks with an action plan and timetable for the completion of the additional work. We plan to share your responses with the reviewers and then issue a binding recommendation.

The enthusiasm for the new technique presented in the paper was high. However, the reviewers recommend a clear demonstration that this new technique can be used to discover something new. In other words, the reviewers feel that the paper would be more exciting to a broad readership if results were obtained that would not have been possible with standard analyses already used in the field. The reviewers are confident that the recommended revisions could be done with the authors' existing data sets (i.e. without requiring collection of new data, which would take a long time). Please find the reviewers' specific comments below:

Reviewer #1:

This is a potentially very interesting paper that uses hidden Markov models to uncover structure in hippocampal place cell ensembles during sharp wave-related states. The advantage of this method, compared to existing methods, is that place cell sequences can be identified without the use of a place cell training data set from active running behaviors. This could allow for the identification of sequences that represent non-spatial memories or sequences from single trials. However, the paper is written in a way that may not be readily understandable by a general readership. My specific comments are as follows:

1) I am afraid that the HMM approach is not explained in a way that is entirely clear to a general audience. For example, the authors write, "the unobserved latent variable represents the temporal evolution of a memory trace which is coherent across the CA1 ensemble" (subsection “Learning hidden Markov models from PBE data”). What does this mean exactly? Similarly, the meaning of "the probability that the CA1 ensemble will transition from a start state to a destination state in the next time bin" is unclear. It seems like a state basically corresponds to a representation of a location, but this is not really explicitly explained. I did not understand the cartoon and its explanation in Figure 1A. Does this represent sequences? Do the colors correspond to the place cells shown? In Figure 2, I did not understand what was meant by "nodes on the same radius correspond to the same state". Also, in Figure 2, I did not understand the difference between panels C and E, and between panels D and F. The authors should probably better explain the measures from graph theory since they will probably be unknown to many readers. In subsection “Ordering states for visualization”, the authors refer to the state with the highest initial probability, but I did not understand what this meant exactly. The readers use HMM jargon like "the forward-backward algorithm" that may need to be briefly explained.

2) It seems concerning that none of the data from rat 3 seems to show a difference between the actual data and the surrogate data. (Figure 1—figure supplement 1, Figure 2—figure supplement 1).

3) It would be useful to have more information about the human scorers. Who were they? How were they trained? The authors write (subsection “Human scoring and detection comparison”) that an event was designated as replay if it was labeled as "excellent" or "good" by a majority of scorers. Do they mean just more than half of the scorers?

Reviewer #2:

Maboudi et al., propose the use of Hidden Markov Models to represent multi-neuron activity patterns in the hippocampal subfield CA1. The model is trained from activity in Population Bursts Events (Likely to a large extent coinciding with sharp wave ripple events). It is shown that HMM can successfully capture at least part of the statistical structure of the data, and while they are trained on spontaneous activity, they can recover the spatial responses observed during active behavior. The method has several attractive features, most importantly, that it detects patterns in an unsupervised fashion. In the literature, spontaneous activity in the hippocampus has been typically analyzed in the context of "replay" and making use of Bayesian decoders trained on the spatial responses of the neurons. While this approach has been quite successful, it also restricts analysis to patterns that are representative of the behavioral situation (spatial environment etc.) that has been used to train the decoder itself. Whatever information is carried by the activity, but not covered by the decoder, goes undetected. Unsupervised methods hold the promise of uncovering structure in a template free way.

I am quite sympathetic with the approach, but I think that it needs a better demonstration of its success and usefulness. I am fairly confident this can be done based on the data the authors use here, and to a large extent the same methods. My main points:

1) The used surrogate data methods (essentially column-wise and row-wise shuffling), is the intuitive one, and probably correct. We see that the sparseness of the model trained from real data is higher than for the shuffled data. However, the effects (Figure 2C-F) seem to be very small. One is then left wondering about what the model is actually doing. For example, one could imagine this pattern of results may emerge in the extreme sparsity case, when each state only represents activation of one neuron. In that case, shuffling would clearly leave the sparsity of the observation matrix unchanged. Why the sparsity of the transition matrix doesn't change with shuffling is more puzzling and should be investigated

2) The PBE derived model recovers the spatial response properties of the ensemble quite well. I am a bit puzzled about how this reflects the known statistics of hippocampal activity though. We know that on a given track (e.g. Figure 3), about 30% of CA1 neurons fire, while during SWRs most neurons activate. Then, how is it possible that all states derived by PBE are activated on the track? wouldn't we expect only a subset to do so?

3) I am a bit puzzled by the congruency measure. We see in Figure 4 that the model with the shuffled transition matrix yields lower likelihood scores, however in the examples shown in Figure 4B we see that the sequential structure is by and large preserved even in the shuffled model. This seems to suggest that the HMM optimization (forward/backward algorithm) is dominated by the observation probabilities, and the transition matrix is less important. But then one should ask what of the essential structure of the data is the transition matrix capturing.

4) I understand that organizing a "human scorers team" was probably a huge effort. Yet I am not sure that this is the best comparison for the analytical procedure, given its strongly subjective nature.

5) A more general question: The bottom line of the paper is that the outcome of PBE/HMM parallels the outcome of Bayesian decoding. Yet what would be most interesting is what you may get *in addition* to what you get from Bayesian methods. In particular, focusing on the events that are not detected by Bayesian decoding but are detected by PBE/HMM may be interesting, and give more detail on their time distribution, structure etc.

Reviewer #3:

The main contribution of this paper is the Introduction of a standard analysis framework (hidden Markov models – HMMs) to the study of place cells in the hippocampus during population burst events (PBEs) when the animal is *not* moving and reply events are believed to be happening. In the standard analysis techniques, behavioral data (e.g. location) is used during navigation to characterize tuning properties of individual neurons and then activity sequences during PBE events is viewed through this tuning to determine if there is replay activity consistent with the measured tuning properties. In the proposal of this manuscript, the HMM analysis technique can be used only on the PBE events and the resulting latent states appear to discover the same localized spatial tuning as what has been found during active navigation. These result show that a completely unsupervised method (i.e. NOT using behavioral location data) can be used to find structure and assess questions about replay, demonstrating that the structure of activity during PBEs is strong enough to discover in a data-driven way.

In my opinion, the technique is promising and has potential to uncover very interesting information coding principles and test hypotheses. As a methods paper, I think it is a strong contribution to the literature. For *eLife*, I am unclear whether it is a good fit because I find the current draft lacking in the articulation of the scientific discovery that the technique enables. What have we learned about the system by doing the analysis in the paper? While I find the analysis interesting, I'm having trouble understanding what we've learned that has significant impact on our understanding. Similarly, for the future, the manuscript points toward some novel analysis methods that this approach would allow but never quite makes a compelling case for what we would learn from it. This is a good paper and should be published, and I am open to be convinced that it should be published in *eLife*. But, I think the manuscript needs to make a more compelling case for what we can learn from the data being analyzed in this way that we cannot learn otherwise.

- Some of the language in the Introduction regarding the motivation for the study is vague and unclear. What does it mean for the "fundamental mode" of the hippocampus to be "PBE sequences"? This seems like a statement trying to make a big claim that their study could support, but it's unclear and doesn't quite make the connection for me.

- In the Results section, I think you could be a little more clear on the details like 1) what is the latent state representing (position), what is the prior, and what is the mapping to Λ (e.g., the tuning curve from state to firing rate). These come up in the Materials and methods but giving a little hint here would help make things more clear.

- I find some of the figure captions too abbreviated to actually understand what's going on in the figures. For example, in Figure 1A, I don't really understand the plot on the bottom left.

- In subsection “What do the learned model parameters tell us about PBEs?”, the latent states are used to estimate a likelihood for position on the track. The comparison is made with the shuffled data, but there is no comparison to the decoding using the standard technique (i.e., place cell tuning curves). I would expect the standard decoding to do better (since it's developed with a supervised learning approach), but it would be good to show the comparison and discuss.

- In subsection “What do the learned model parameters tell us about PBEs?” where the manuscript discusses assessing the patterns of activity for congruence, I think I am confused about the approach. The method is described as comparing against a single null model. As far as I understand, these results would depend entirely on the specific form of the null model that is the baseline for comparison. Why is this an acceptable approach? It seems so dependent on that arbitrary decision that I don't understand how the results are interpretable as a generalized consistency metric.

- There were some punctuation and wording issues that affected clarity slightly. Subsection “HMM-congruent PBEs capture sequence replay”: "between model quality or the number of PBEs and Bayesian Decoding". Subsection “Modeling internally generated activity during open 1eld behaviour”: "Can an HMM trained on PBEs indeed".

[Editors' note: further revisions were requested prior to acceptance, as described below.]

Thank you for resubmitting your work entitled "Uncovering temporal structure in hippocampal output patterns" for further consideration at *eLife*. Your revised article has been favorably evaluated by Michael Frank (Senior Editor) and a Reviewing Editor.

The manuscript has been greatly improved but there are some remaining issues that need to be addressed before acceptance, as outlined below.

In particular, the authors' analyses of events that were non-significant with the Bayesian approach and demonstration that many of them correspond to replayed representations of a different, previously explored environment (i.e., new Figure 7) satisfactorily address the major concerns raised during the first round of review.

However, the description of Figure 7 is not entirely clear. For example, the x-axes in panels D and I are labeled "time", but the color bar presumably corresponding to space is also shown below the x-axis. What is this supposed to indicate? Also, panels E-G and J-L need a more thorough explanation. The figure legend states that panel e (bottom) shows decoded positions using E2, which is the local environment, yet the top of panel e presumably corresponds to the remote environment (i.e. E1). This is confusing. The descriptions of Figure 7A-B in the Results section and the figure legend are also not clear. The text states that the model quality decreased rapidly when remote replay events were removed, and that model quality decreased slowly when local replay events were removed. But, the figure appears to show the opposite. The light blue line (local removed) decreases more rapidly than the green line (remote removed). Also, are the top two slowest high quality decay sessions indicated in panel B the ones that were used to find remote replay events? It isn't clear what panel B is supposed to show- it should be explained clearly in the text and in the figure legend.

There is also insufficient detail provided about Figure 8. How do these figures relate to the object-location memory task? Couldn't these sleep sequences simply have been detected from exploration of the 60 x 60 enclosure regardless of object-location memory? And wouldn't they also have been potentially detected with Bayesian decoding? The authors claim in subsection “Extra-spatial Information” that the HMM can uncover sequential activity that is not readily detectable with the Bayesian approach, but it is unclear how Figure 8 shows this. In any case, Figure 8 is not really necessary as long as Figure 7 is explained clearly.

A few minor issues also remain to be clarified. In subsection “Awake population burst events”, the authors state that similar results were obtained when a theta detection approach was used, but it is unclear where this approach is shown or described. Also, it is unclear to what "these" refers. Also, in Figure 1B, the "pyramidal cells (non place)" label should be changed to "putative pyramidal cells (non place)". Also, the difference between the mean departure Gini and mean Λ row Gini in Figure 6E-F should be explained more clearly for a general readership. Finally, as was raised in the initial review, the authors should explicitly state in the first two paragraphs of the Materials and methods section that the Diba and Buzsaki and Pfeiffer and Foster datasets were recorded previously and used in other publications. Otherwise, readers may misunderstand and assume that methods were adopted from these papers, but that similar data were collected again for the present study.

---

## [Author Response]

The enthusiasm for the new technique presented in the paper was high. However, the reviewers recommend a clear demonstration that this new technique can be used to discover something new. In other words, the reviewers feel that the paper would be more exciting to a broad readership if results were obtained that would not have been possible with standard analyses already used in the field. The reviewers are confident that the recommended revisions could be done with the authors' existing data sets (i.e. without requiring collection of new data, which would take a long time).

In addition to revising the text to better explain the methodology to a lay reader, we have performed a series of new analyses on our existing data as well as two additional datasets that highlight the novelty and power of our approach. Inspired by the second reviewer’s suggestions, we removed events that were detected as significant using Bayesian replay and retrained the hidden Markov model (HMM) on the remaining (non-significant) events. We note, as suggested by the reviewer, that this approach could not even be attempted with a Bayesian decoding approach. Interestingly, we found that in some sessions the HMM still captured a large of amount of the structure in the remaining population burst events (PBEs) (new Figure 7A). We conjectured that these events may correspond to replay of a remote environment. By analyzing a dataset where the recording spanned behavior of an animal in two separate environments, we confirmed that indeed many HMM-congruent events that are found as non-significant with the Bayesian approach correspond to replay of the remote environment. As a final test, we applied the HMM to PBEs in sleep following the exploration phase of a hippocampus-dependent object-location task (Figure 8) and found that we could detect a significant number of HMM-congruent events in post-task sleep, without reference to behavior. We believe these new results significantly elevate the impact of our report and we thank the reviewers for their suggestions in our acknowledgements.

Reviewer #1:This is a potentially very interesting paper that uses hidden Markov models to uncover structure in hippocampal place cell ensembles during sharp wave-related states. The advantage of this method, compared to existing methods, is that place cell sequences can be identified without the use of a place cell training data set from active running behaviors. This could allow for the identification of sequences that represent non-spatial memories or sequences from single trials. However, the paper is written in a way that may not be readily understandable by a general readership. My specific comments are as follows:1) I am afraid that the HMM approach is not explained in a way that is entirely clear to a general audience. For example, the authors write, "the unobserved latent variable represents the temporal evolution of a memory trace which is coherent across the CA1 ensemble" (subsection “Learning hidden Markov models from PBE data”). What does this mean exactly? Similarly, the meaning of "the probability that the CA1 ensemble will transition from a start state to a destination state in the next time bin" is unclear. It seems like a state basically corresponds to a representation of a location, but this is not really explicitly explained.

We have now revised the manuscript with an eye towards further clarity and well-explained terminology. With regards to these specific examples, we now clarify that a HMM is a model of ensemble firing (coactive neurons corresponding to latent states) and how these ensembles change over time. While we do find that states correspond to location, that is only because location is the organizing principle of the activity in the task under consideration. In a different task, states might very well correspond to time, frequency space, or some more complex arbitrary manifold.

The specific lines mentioned by the reviewer have been eliminated or revised and the beginning of the Results section now provides a better introduction to the HMM methodology.

"Hidden Markov models have been very fruitfully used to understand sequentially structured data in a variety of contexts. A hidden Markov model captures information about data in two ways. […] Given the role of the hippocampus in memory, in our HMM, the unobserved latent variable presumably corresponds to the temporal evolution of a memory trace that is represented by co-active ensembles of CA1 and CA3 neurons. The full model will correspond to the structure which connects all the memory traces activated during PBEs."

I did not understand the cartoon and its explanation in Figure 1A. Does this represent sequences? Do the colors correspond to the place cells shown?

The cartoon illustration has been removed from Figure 1 and has been replaced by a latent-state-space-view that hopefully helps readers understand the salient parts of the HMM approach better than the cartoon did.

In Figure 2, I did not understand what was meant by "nodes on the same radius correspond to the same state".

We have now clarified that the inner and outer rings represent the same nodes – this makes it easier to visualize the self-transitions (starting and ending in the same state) and the transitions to the next node. In the caption, we now state:

"The nodes correspond to states (progressing clockwise, starting at the top). The weights of the edges are proportional to the transition probabilities between the nodes (states). The transition probabilities from state i to every other state except i+1 are shown in the interior of the graph, whereas for clarity transition probabilities from state i to itself, as well as to neighboring state i+1 are shown between the inner and outer rings of nodes (the nodes on the inner and outer rings represent the same states)."

Also, in Figure 2, I did not understand the difference between panels C and E, and between panels D and F.

We have revised the captions of the figure to clarify that panels C and E are the transition matrices (state dynamics) for the example sessions and all sessions, respectively, and D and F are the observation models (firing rate / state) for the example session and all sessions, respectively.

The authors should probably better explain the measures from graph theory since they will probably be unknown to many readers.

In the Materials and methods section, we now expand on the Gini coefficient when it is introduced:

"Sparsity of the transitions from individual states (departure sparsity) was measured by calculating the Gini coefficient of corresponding rows of the transition matrix (Hurley and Rickard, 2009). The Gini coefficient is a measure of how variable the values of this probability distribution are, with equality across states corresponding to a coefficient of zero (minimal sparsity), and a singular distribution with a probability-one transition to a single other state corresponding to a coefficient of one (maximal sparsity). The sparsity of the full transition matrix was calculated by averaging the Gini coefficient across rows."

We also replaced the term “measures from graph theory” in the main text and also referred to Materials and methods section for a more detailed explanation of the search algorithm, “Using a graph search algorithm (see Materials and methods section), we simulated paths through state space generated by […]”

In subsection “Ordering states for visualization”, the authors refer to the state with the highest initial probability, but I did not understand what this meant exactly.

The Initial State probability is one of the parameter estimated during model learning, along with the Observation Model and State Transition Probabilities. It describes the probability that a given state will be the beginning of a sequence. We now explicitly state:

“The parameters of our model that are fit to data include the observation model (the cluster descriptions or predicted activity of each excitatory neuron within the CA1/CA3 ensemble for a given state), the state transition model (the probability that the CA1/CA3 ensemble will transition from a start state to a destination state in the next time bin), and the initial state distribution (the probability for sequences to start in each given state). In prior work using HMMs to model neural activity, a variety of statistical distributions have been used to characterize ensemble firing during a specific state (the observation model.”

The readers use HMM jargon like "the forward-backward algorithm" that may need to be briefly explained.

We have revised the text throughout to replace jargon with more readily understandable explanations. e.g.

“To observe the evolution of the latent states during behavior, we used our model to determine the most likely sequence of latent states corresponding to the neural activity observed in 100 ms bins during epochs that displayed strong theta oscillations (exclusive of PBEs) when rats were running (speed > 10 cm/s; see Materials and methods section).”

And in the Materials and methods section, we have expanded the description to say:

"For a given PBE (i.e. observation sequence) with K bins, we use the “forward-backward algorithm” (Rabiner, 1989) to calculate both the probability distribution of the latent state for each time bin, Pr(*q_t_*|*O*_1_,…,*O*_t_,…,*O_K_*). For a particular time bin, *t*, in a given sequence, the forward-backward algorithm allows information from all observation bins, previous and subsequent, to affect this state probability distribution (as well the observation bin at time *t*). The forward-backward algorithm also can efficiently calculate, and the “score”, or likelihood of the complete sequence, Pr(*O*_1_,…,*O_K_*)."

2) It seems concerning that none of the data from rat 3 seems to show a difference between the actual data and the surrogate data. (Figure 1—figure supplement 1, Figure 2—figure supplement 1).

The models from rat 3 were indeed generally of lower quality largely because of a lower number of PBEs (Figure 5—figure supplement 1C). However, we would point out that the sessions from this animal still yielded transition matrices that were sparser than shuffles (significance stars in Figure 2—figure supplement 1).

3) It would be useful to have more information about the human scorers. Who were they? How were they trained? The authors write (subsection “Human scoring and detection comparison”) that an event was designated as replay if it was labeled as "excellent" or "good" by a majority of scorers. Do they mean just more than half of the scorers?

We now describe the scorers:

"The scorers included six graduate students (including one of the authors) and two undergraduates, all of whom were generally familiar with the concept of hippocampal replay.

And indeed, majority by definition means more than half the scorers:

"An event was then designated as replay if it was thus labeled as “excellent” or “good” by a majority of scorers (ties were labeled as non-replay)."

Reviewer #2:[…] 1) The used surrogate data methods (essentially column-wise and row-wise shuffling), is the intuitive one, and probably correct. We see that the sparseness of the model trained from real data is higher than for the shuffled data. However, the effects (Figure 2C-F) seem to be very small. One is then left wondering about what the model is actually doing. For example, one could imagine this pattern of results may emerge in the extreme sparsity case, when each state only represents activation of one neuron. In that case, shuffling would clearly leave the sparsity of the observation matrix unchanged. Why the sparsity of the transition matrix doesn't change with shuffling is more puzzling, and should be investigated.

We recognize that no single measure captures the entire structure of spatial representations and each shuffle alters this structure in unique ways that preserves some of this structure, as the reviewer has pointed out. For example, because we used a very conservative per-event shuffle in these surrogates, much of the sparsity of the original events was retained. Additionally, the fact that our measure is between 0 and 1 tends to limit the range of the values of sparsity that are found. Thus, differences can appear small even when they are significant. The large variability of session sparsity values across sessions, can further moderate consistent differences. This is why we show the per-session differences in the supplemental figure. Finally, when there is little structure in either the surrogate or actual data, the EM algorithm will tend to converge to solutions in which the observation matrix is degenerate. In these instances when the state likelihoods approach zero, the sparsity measure loses meaning. This reflects the fact that for limited sizes of training data, EM will tend towards solutions which overfit individual observations rather than yielding a transition matrix that would be fully connected. Despite these limitations we show that for the majority of sessions in our dataset the sparsity measure provides a useful measure of the structure of the transition and observation matrices.

2) The PBE derived model recovers the spatial response properties of the ensemble quite well. I am a bit puzzled about how this reflects the known statistics of hippocampal activity though. We know that on a given track (e.g. Figure 3), about 30% of CA1 neurons fire, while during SWRs most neurons activate. Then, how is it possible that all states derived by PBE are activated on the track? wouldn't we expect only a subset to do so?

The reviewer may be thinking of SWRs during deep sleep, whereas here we are reporting on SWRs during quiet waking on the track, during which, in fact, on a subset of neurons are active during SWRs. It is worth noting (and possibly future study) that we observed a bias (~1.5 times) for neurons with place fields vs nonplace field neurons to be activated during PBEs. Nevertheless, a major benefit of an HMM model with unsupervised learning is that with sufficient training data the learned model is often insensitive to variability that non-maze-active neurons might represent. Nevertheless, inspired by the reviewer’s comments, we have performed new analyses where we investigate events which do not decode well to trajectories in the animal’s current environment (Figure 7). Remarkably, we find that some of these events decode to trajectories in a previously-experienced remote environment.

3) I am a bit puzzled by the congruency measure. We see in Figure 4 that the model with the shuffled transition matrix yields lower likelihood scores, however in the examples shown in Figure 4B we see that the sequential structure is by and large preserved even in the shuffled model. This seems to suggest that the HMM optimization (forward/backward algorithm) is dominated by the observation probabilities, and the transition matrix is less important. But then one should ask what of the essential structure of the data is the transition matrix capturing

The reviewer is correct that the transition matrix shuffle that we have implemented by design retains much of the structure of the original data that is captured by the model. It is worth noting that we shuffle only across rows of the transition matrix, but not columns, and we exclude elements on the diagonal Our aim with this shuffle is strictly to provide a test comparison against the sequential component of the original model. To make this point better and avoid confusion, we now clarify in the text:

“Note that while we describe this as HMM-congruence, we have maintained the diagonal of the transition matrix, which specifically selects against PBEs which might be model-congruent by maintaining a single state over many time bins. In reality there are other dimensions of the HMM that we could assess congruence against, for example the observation model, the initial state distribution, or combinations of these and the transition matrix. In comparing against Bayesian decoding, our current definition seemed most appropriate for sequence detection, but we can imagine future studies expanding on our approach.”

4) I understand that organizing a "human scorers team" was probably a huge effort. Yet I am not sure that this is the best comparison for the analytical procedure, given its strongly subjective nature.

While some subjectivity is inherent in human scoring of replay, we argue here that methods and thresholds used in different studies also inherently involve subjective criterion. This inherent subjectivity is precisely what we want to reveal for readers who may have the sense that patterns which exceed a 95% threshold against a shuffle distribution are very different from those at 94.5%. We do not make the claim that human scorers provide “the best comparison” for replay detection. The point of showing results from human scorers is to contradict the sense in the literature that a binary distinction between replay sequences and non-replay can be easily identified in the *a posteriori* likelihood of Bayesian decoding.

5) A more general question: The bottom line of the paper is that the outcome of PBE/HMM parallels the outcome of Bayesian decoding. Yet what would be most interesting is what you may get in addition to what you get from Bayesian methods. In particular, focusing on the events that are not detected by Bayesian decoding but are detected by PBE/HMM may be interesting, and give more detail on their time distribution, structure etc.

We appreciate this suggestion of the reviewer, as it led us to add an additional section in the revised paper. Specifically, we investigate how our model quality measure changes as we remove Bayesian decoding significant sequences from the training data. We note that such an analysis is not conceivable using the Bayesian methodology. For many of our data sets, the model quality drops very quickly, which suggests to us that the PBEs in these data are largely encoding information about the local environment. However, in other data sets, including in sessions during which remote replay had previously been reported, we find that the model quality changes more gradually when Bayesian replay events are dropped. Remarkably, when we investigate specific HMM-congruent events that fail to pass the threshold for Bayesian replay in these data, we find that they correspond to remote replay of the previously-experienced environment. Thus, we now suggest that combining information from the Bayesian-decoding approach and the HMM-congruence measure in this way would be a way to specifically look for extra-spatial or remote replay.

Reviewer #3:[…] In my opinion, the technique is promising and has potential to uncover very interesting information coding principles and test hypotheses. As a methods paper, I think it is a strong contribution to the literature. For eLife, I am unclear whether it is a good fit because I find the current draft lacking in the articulation of the scientific discovery that the technique enables. What have we learned about the system by doing the analysis in the paper? While I find the analysis interesting, I'm having trouble understanding what we've learned that has significant impact on our understanding. Similarly, for the future, the manuscript points toward some novel analysis methods that this approach would allow but never quite makes a compelling case for what we would learn from it. This is a good paper and should be published, and I am open to be convinced that it should be published in eLife. But, I think the manuscript needs to make a more compelling case for what we can learn from the data being analyzed in this way that we cannot learn otherwise.

We have revised the manuscript and performed a number of new analyses to highlight the novel benefits of our approach. First, we would argue that no previous analysis has revealed that PBEs contain enough information to form a complete map of the environment. Inspired by the second reviewer, we have also now shown how we can combine Bayesian decoding and HMM-congruence to create a signature for remote replay. While remote replay has been previously reported, no prior approach would allow for remote replays to be identified without behavioral data. Finally, we also now note how the distribution of HMM-congruence can be used to indicate the presence of consistent, ordered sequential activity in PBE data. Bayesian decoding maps between behavior tracking, neural activity during behavior, and neural activity during PBEs. By performing unsupervised learning on the PBEs, we reveal that we can infer important useful structure without these other data, and when we have them, can use them to further illuminate. We believe this is a powerful and useful technique that may have significant, widespread potential and will be of interest to many *eLife* readers.

- Some of the language in the Introduction regarding the motivation for the study is vague and unclear. What does it mean for the "fundamental mode" of the hippocampus to be "PBE sequences"? This seems like a statement trying to make a big claim that their study could support, but it's unclear and doesn't quite make the connection for me.

We have revised the manuscript throughout to make it more precise and attempted to remove any unsupported claims. The motivation of our study was to figure out if we could infer the structure of hippocampal activity only from PBEs. We believe that we have presented substantial evidence that this is possible. We agree that calling PBEs a "fundamental mode" would be self-aggrandizing. However, ours is a question rather than a claim and is meant to provoke further consideration; if a reader believes that "place cell activity during exploration is a fundamental mode of the hippocampus", we hope to provoke them to ask whether it is possible that PBEs might also be and to consider what evidence would demonstrate this.

- In the Results section, I think you could be a little more clear on the details like (1) what is the latent state representing (position), what is the prior, and what is the mapping to Λ (e.g., the tuning curve from state to firing rate). These come up in the Materials and methods but giving a little hint here would help make things more clear.

The text and Materials and methods section have been revised for improved clarity. We note, however, that the HMM is agnostic as to whether the latent state represents position or another construct (e.g. time or memory). In the Results section we now provide additional description:

“Hidden Markov models have been very fruitfully used to understand sequentially structured data in a variety of contexts. A hidden Markov model captures information about data in two ways. […] In prior work using HMMS to model neural activity, a variety of statistical distributions have been used to characterize ensemble firing during a specific state (the observation model.”

- I find some of the figure captions too abbreviated to actually understand what's going on in the figures. For example, in Figure 1A, I don't really understand the plot on the bottom left.

The cartoon illustration has been removed from Figure 1 and has been replaced by a latent-state-space-view that hopefully helps readers understand the salient parts of the HMM approach better than the cartoon did.

- In subsection “What do the learned model parameters tell us about PBEs?”, the latent states are used to estimate a likelihood for position on the track. The comparison is made with the shuffled data, but there is no comparison to the decoding using the standard technique (i.e., place cell tuning curves). I would expect the standard decoding to do better (since it's developed with a supervised learning approach), but it would be good to show the comparison and discuss.

We have added median Bayesian decoding error to the first panel of the Figure 3—figure supplement 1. At the end of that section, we have added the text:

"Note that decoding into the latent space and then mapping to position resulted in slightly higher error than simply performing Bayesian decoding on the neural activity during behavior. This suggests that the latent space we learn from PBEs may not capture all the information about space that is present in hippocampal activity during behavior, though this may reflect the limited number of PBEs from which we can learn."

- In subsection “What do the learned model parameters tell us about PBEs?” where the manuscript discusses assessing the patterns of activity for congruence, I think I am confused about the approach. The method is described as comparing against a single null model. As far as I understand, these results would depend entirely on the specific form of the null model that is the baseline for comparison. Why is this an acceptable approach? It seems so dependent on that arbitrary decision that I don't understand how the results are interpretable as a generalized consistency metric.

The reviewer makes a very insightful remark. We have added additional text to the Materials and methods section to clarify:

"Note that while we describe this as HMM-congruence, we have maintained the diagonal of the transition matrix, which specifically selects against PBEs which might be model-congruent by representing a single state for many time bins. In reality there are other dimensions of the HMM that we could assess congruence against, for example the observation model, the initial state distribution, or combinations of these and the transition matrix. In comparing against Bayesian decoding, our current definition seemed most appropriate, but we can imagine future studies expanding on our approach."

- There were some punctuation and wording issues that affected clarity slightly. Subsection “HMM-congruent PBEs capture sequence replay”: "between model quality or the number of PBEs and Bayesian Decoding". Subsection “Modeling internally generated activity during open 1eld behaviour”: "Can an HMM trained on PBEs indeed"

The first sentence has been rephrased:

"Not surprisingly, the performance of Bayesian decoding relative to human scorers was independent of the quality of the HMM model or the number of PBEs as the place field model is learned from ensemble neural activity during running."

And the second sentence has been rephrased:

"Do the latent states learned from Can an HMM trained on PBEs indeed capture spatial information about in a 2D environment?"

[Editors' note: further revisions were requested prior to acceptance, as described below.]

The manuscript has been greatly improved but there are some remaining issues that need to be addressed before acceptance, as outlined below.In particular, the authors' analyses of events that were non-significant with the Bayesian approach and demonstration that many of them correspond to replayed representations of a different, previously explored environment (i.e., new Figure 7) satisfactorily address the major concerns raised during the first round of review.However, the description of Figure 7 is not entirely clear. For example, the x-axes in panels D and I are labeled "time", but the color bar presumably corresponding to space is also shown below the x-axis. What is this supposed to indicate?

Color in panel (c) and the bottom of (e) corresponds to time (this copies the original Karlsson and Frank scheme). To make things clearer, we’ve split panel (e) into (e)/(f) and updated the caption for panel (f) to indicate that the color is the same as in (c). This allows us to remove the color bar from the middle of (d) which was understandably confusing.

Also, panels E-G and J-L need a more thorough explanation. The figure legend states that panel e (bottom) shows decoded positions using E2, which is the local environment, yet the top of panel e presumably corresponds to the remote environment (i.e., E1). This is confusing.

Alongside the lettering changes, we’ve updated the figure caption to better explain this figure:

“c–n. Two example HMM-congruent but not Bayesian-significant events from the W maze session are depicted to highlight the fact that congruence can correspond to remote replay. c. Spikes during ripple with local place cells highlighted. d. In this event, the Bayesian estimates of position using linearized local place fields have jumpy, multi-modal *a posteriori* distributions over space in 1D and 2D (2D depicts distribution modes and time is denoted in color). e. The replay score relative to the shuffle distribution indicates that this event is not Bayesian significant. e. Latent state probabilities decoded using the HMM trained on PBEs show sequential structure (grayscale intensity corresponds to probability). f. The distribution of HMM-score shuffles indicate the ripple event is HMM-congruent. h. Bayesian decoding using the remote environment indicate that the sample event is a remote replay. Note that in a typical experiment, only the local place fields would be available. i–n. Same as c–h, but for a different ripple event.”

The descriptions of Figure 7A-B in the Results section and the figure legend are also not clear. The text states that the model quality decreased rapidly when remote replay events were removed, and that model quality decreased slowly when local replay events were removed. But, the figure appears to show the opposite. The light blue line (local removed) decreases more rapidly than the green line (remote removed).

We apologize for this error. Indeed, as understood by the editors, the caption and text were correct, but the legend text on the figure itself mis-identified the lines and which events were being removed. This has now been corrected in the figure.

Also, are the top two slowest high quality decay sessions indicated in panel B the ones that were used to find remote replay events? It isn't clear what panel B is supposed to show- it should be explained clearly in the text and in the figure legend.

The flow of text was indeed confusing. We’ve removed the red dot from the figure, and specifically discussed the blue line and blue dot in the figure caption and the text. From the text:

“Captions: The blue line identifies a session in which model quality drops more slowly, indicating the potential presence of extra-spatial information. The reduction in session quality for a W maze experiment with known extra-spatial information is even slower (green) […] The blue dot indicates the outlier session from panel a with potential extra-spatial information: this session shows high decoding error combined with high session quality.

Text: Intriguingly, however, in at least one outlier session where model quality decreased slowly with refinement (blue line, Figure 7A), the initial model quality was still high, and we further noted that position decoding using lsPFs yielded relatively high error (blue dot, Figure 7B). Thus, we wondered whether this and similar sessions might have contained non-local or extra-spatial PBEs that were captured by the HMM.”

There is also insufficient detail provided about Figure 8. How do these figures relate to the object-location memory task? Couldn't these sleep sequences simply have been detected from exploration of the 60 x 60 enclosure regardless of object-location memory? And wouldn't they also have been potentially detected with Bayesian decoding? The authors claim in subsection “Extra-spatial Information” that the HMM can uncover sequential activity that is not readily detectable with the Bayesian approach, but it is unclear how Figure 8 shows this. In any case, Figure 8 is not really necessary as long as Figure 7 is explained clearly.

We agree with this assessment and have revised the text to better avoid unjustified claims. We’ve removed the phrase “not readily detectable with the Bayesian approach”. eloThe motivation for this analysis is also better explained—to examine the potential for HMM is uncovering sequence structure during sleep (rather than only waking as elsewhere in our manuscript), without reference to behavior. We’ve streamlined the text to emphasize model quality assessment and removed the second panel from the figure. If the editor prefers, we are also willing to remove this figure entirely.

A few minor issues also remain to be clarified. In subsection “Awake population burst events”, the authors state that similar results were obtained when a theta detection approach was used, but it is unclear where this approach is shown or described.

For the sake of space, we did not show these results. We’ve updated the text to state this.

“While we identified active behavior using a speed criterion, we found similar results when we instead used a theta-state detection approach (not shown).”

Also, in subsection “Awake population burst events”, it is unclear to what "these" refers.

We changed the text to state:

“We found that inactive PBEs occupied an average of 1.8% of the periods during which animals were on the linear track.”

Also, in Figure 1B, the "pyramidal cells (non place)" label should be changed to "putative pyramidal cells (non place)".

This has been updated.

Also, the difference between the mean departure Gini and mean Λ row Gini in Figure 6E-F should be explained more clearly for a general readership.

We modified the earlier text to better explain the intuition behind Gini coefficients:

“First, we investigated the sparsity of the transition matrices using the Gini coefficient (see Materials and methods and Figure 2—figure supplement 1). A higher Gini coefficient corresponds to higher sparsity. Strikingly, the actual data yielded models in which the state transition matrix was sparser than in each of the surrogate counterparts (p < 0.001, Figure 2C), reflecting that each state transitions only to a few other states. Thus, intricate yet reliable details are captured by the HMM. Next, we quantified the sparsity of the observation model. We found that actual data yielded mean firing rates which were highly sparse (Figure 2D), indicating that individual neurons were likely to be active during only a small fraction of the states.”

And, we changed the labels in Figure 6 to match Figure 2, and updated the caption to say:

“Comparing the sparsity of the transition matrices (mean Gini coefficient of the departure probabilities) between the linear track and open field reveals that, as expected, over the sessions we observed, the open field is significantly *less sparse* (p<0.001), since the environment is less constrained. f. In contrast, there is not a significant difference between the sparsity of the observation model (mean Gini coefficient of the rows) between the linear track and the open field.”

Finally, as was raised in the initial review, the authors should explicitly state in the first two paragraphs of the Materials and methods section that the Diba and Buzsaki and Pfeiffer and Foster datasets were recorded previously and used in other publications. Otherwise, readers may misunderstand and assume that methods were adopted from these papers, but that similar data were collected again for the present study.

We now say “As previously reported using these same data” in reference to Diba and Buzsaki, and “Briefly, as was previously reported using this dataset” in reference to Pfeiffer and Foster.